# Biological Effects of Quinolones: A Family of Broad-Spectrum Antimicrobial Agents

**DOI:** 10.3390/molecules26237153

**Published:** 2021-11-25

**Authors:** Ana R. Millanao, Aracely Y. Mora, Nicolás A. Villagra, Sergio A. Bucarey, Alejandro A. Hidalgo

**Affiliations:** 1Facultad de Ciencias, Instituto de Farmacia, Universidad Austral de Chile, Valdivia 5090000, Chile; anamillanao@uach.cl; 2Programa de Doctorado en Bioquímica, Universidad de Chile, Santiago 8380544, Chile; aracely.mora@ug.uchile.cl; 3Escuela de Tecnología Médica, Universidad Andres Bello, Santiago 8370071, Chile; nicolas.villagra@unab.cl; 4Facultad de Ciencias Veterinarias y Pecuarias, Universidad de Chile, Santiago 8820808, Chile; sbucarey@uchile.cl; 5Escuela de Química y Farmacia, Universidad Andres Bello, Santiago 8370071, Chile

**Keywords:** quinolones, fluoroquinolones, antimicrobial, antibiotics, ciprofloxacin

## Abstract

Broad antibacterial spectrum, high oral bioavailability and excellent tissue penetration combined with safety and few, yet rare, unwanted effects, have made the quinolones class of antimicrobials one of the most used in inpatients and outpatients. Initially discovered during the search for improved chloroquine-derivative molecules with increased anti-malarial activity, today the quinolones, intended as antimicrobials, comprehend four generations that progressively have been extending antimicrobial spectrum and clinical use. The quinolone class of antimicrobials exerts its antimicrobial actions through inhibiting DNA gyrase and Topoisomerase IV that in turn inhibits synthesis of DNA and RNA. Good distribution through different tissues and organs to treat Gram-positive and Gram-negative bacteria have made quinolones a good choice to treat disease in both humans and animals. The extensive use of quinolones, in both human health and in the veterinary field, has induced a rise of resistance and menace with leaving the quinolones family ineffective to treat infections. This review revises the evolution of quinolones structures, biological activity, and the clinical importance of this evolving family. Next, updated information regarding the mechanism of antimicrobial activity is revised. The veterinary use of quinolones in animal productions is also considered for its environmental role in spreading resistance. Finally, considerations for the use of quinolones in human and veterinary medicine are discussed.

## 1. Introduction

Transmissible diseases kill an estimate of over 15 million people every year worldwide, with many deaths caused by bacterial infections. Therefore, efficient antimicrobial therapies are crucial to combat infectious diseases. Discovered as a by-product, in the search for improved synthesis of the anti-malarial chloroquine, quinolones are one of the most important families of antimicrobial agents, extensively used to combat bacterial infections. Since the origin of quinolones is fully synthetic, they represent a truly synthetic antimicrobial, in opposition to antibiotics which can trace their origin to a natural product. Therefore, through this text we will refer to quinolones as antimicrobials or antibacterials. However, occasionally the antibiotic designation can also be used. Though the last six decades the quinolones have become one of the most successful antimicrobials from many points of view including but not limited to spectrum, administrations, and tissue distribution. In addition, their versatile chemistry allowed early improvements to produce the quinolones in use today and the future improvements. Today it is known that molecules in the quinolone family are also present as natural products of plant and bacterial origin, and their potency has also been tested as antimicrobials at the experimental level. Since nalidixic acid was introduced in 1962 the family of quinolones has evolved in both efficacy and spectrum. Today more than a dozen quinolones are actively prescribed. One example is ciprofloxacin, a drug widely used to treat a variety of diseases caused by Gram-negative and Gram-positive bacteria, including respiratory tract infections, urinary infections, and several enteric diseases. As it is for other antibiotics, the use of quinolones is not absent of problems. The routine use in the clinic and the sometimes unnecessary use or overuse in animal production farms, has propitiated the appearance of resistance to quinolones in bacteria. 

This review presents an overview of the structures, their evolution and relation to activity. Also, it explores new findings and applications of new naturally occurring and synthetic quinolones and related compounds. Next, the mechanisms of current, but also new biological activities are revised, along with the clinical importance and veterinary use of quinolones. The impact in the environment and up to date mechanisms of resistance are also reviewed before discussing consideration for the use of quinolones to maintain this important family of drugs working against microorganisms.

## 2. Structure and Anti-Bacterial Activity

Few details have been disclosed regarding the origin of the quinolone class of antimicrobials. Although their biological activity was not reported, the structure of basic 3-carboxyquinolones was described back in 1949 by Australian researchers [1,2,3]. Later, a patent was presented in 1957 and published in 1961 by Imperial Chemical Industries, describing quinolones and ascribing anti-bacterial activity. Such quinolones included analogues of nalidixic acid. Only in 1962, Sterling Drugs (later absorbed by Sanofi) published a patent disclosing the anti-bacterial effects of 1,8-naphthyridones, including nalidixic acid. The same year, George Lesher (chemist at Sterling Drugs) published an article detailing the antimicrobials effects of nalidixic acid, which was introduced in the US by Sterling as the first antibacterial in the class of quinolones. It was not until 1986 that during a symposium George Lesher stated that the lead compound was obtained as a by-product during chloroquine synthesis (first molecule in Figure 1). This by-product presented modest antibacterial activity but leads to the synthesis of quinolone analogues, including the naphthyridones and nalidixic acid (second molecular formula in Figure 1) eventually. Most of the published information regarding the origin of quinolones antimicrobials, only cite this last piece of information. However, a detailed review on the discovery and design of first quinolones is presented by Gregory Bisacchi, in his review article entitled “Origins of the Quinolones Class of Antibacterials: An Expanded Discovery Story [4]”. Reading this review is truly recommended for the specialized audience and for the one who likes the history behind the scenes of quinolones invention.

Nalidixic acid was the first in class to be used in the clinic and to-date can still be found in the market in some remote places [5,6], although its use was banned in the United States and European Union. Nalidixic acid presented antimicrobial activity against Gram-negative bacteria and was used mainly to treat urinary tract infections caused by *Escherichia coli* and *Klebsiella pneumoniae* among others; with minimal anti-Gram-positive activity. Renal excretion made nalidixic acid ideal for urinary tract infection, but its use was limited to less complicated infections, due to low potency, high protein binding activity, short half-life and poor bioavailability [7,8]. Perhaps its efficacy, nalidixic acid was gradually replaced with molecules of the new generations of improved quinolones. In the series of three patents by Imperial Chemical Industries during 1961 the structures with additions of a fluorine atom at position six (then called fluoroquinolones, or just FQs, within this text) were described. The introduction of the FQ norfloxacin (fourth molecule in Figure 1), patented in 1978, marked a significant improvement in potency, but also expanded its spectrum, toward Gram-positive bacteria [9]. However, the improvements were not enough, and norfloxacin ultimately failed due to its low blood and fluids concentration and limited Gram-positive activity [10]. The introduction of ciprofloxacin represented a notable improvement in the development of new quinolones and still stands as a useful antimicrobial with excellent tissue distribution and notable activity against Gram-positive bacteria. Ciprofloxacin is the iconic molecule of the second generation of quinolone class of antimicrobials (fifth molecule in Figure 1). Ofloxacin has similar characteristics to ciprofloxacin, but it was only after the development of protocols to purify its left-handed isomer levofloxacin, that it presented new improvements in potency and broad spectrum toward Gram-positive. These compounds found good clinical use to treat common infections of the respiratory and the urinary systems, in inpatients and outpatients, and belong to the WHO-list of essential medicines, with ciprofloxacin and levofloxacin among the most prescribed drugs [5]. The most recent quinolones introduced to clinical use are third and fourth generation levofloxacin and moxifloxacin. These quinolones possess improved Gram-positive antibacterial activities. As the family of quinolones add new members, the new molecules present improved activity against anaerobic bacteria. For example, early during development of quinolones, ciprofloxacin and ofloxacin were described as only active against *Propionibacterium acnes* and some *Clostridium perfringens* strains [11,12,13], among Gram-positive bacteria.

**Figure 1 molecules-26-07153-f001:**
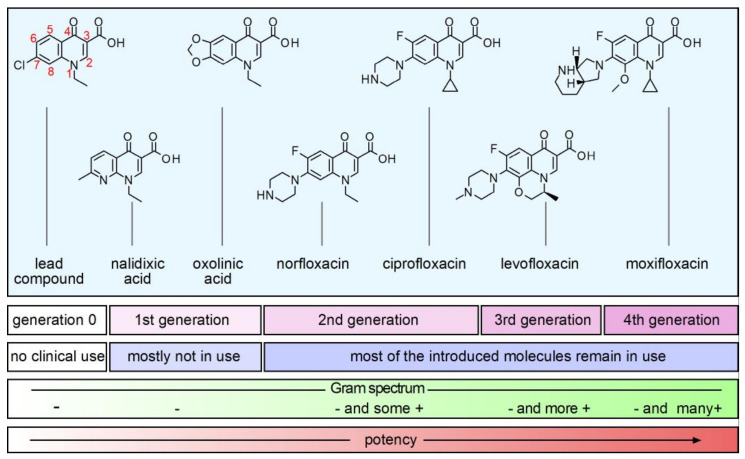
Overview of structures, classification, and relevant clinical aspects of the most important quinolones. Representation of potency is qualitative. The quinolones were classified according to time of discovery or marketed date [14,15]. Chemical formulae were prepared using a free version of ChemSketch (ACD/Labs, Toronto, ON, Canada).

At the experimental level, several quinolones have presented strong anaerobic bacterial activity. One such example is clinafloxacin which presents strong antimicrobial activity in a panel of anaerobes including *Clostridium* (now called *Clostridoides*) and *Bacteroides* among other anaerobes [16]. However, up to date, clinafloxacin is not approved for use in humans and its clinical application was halted because of phototoxicity and low blood sugar levels. However, the evidence indicates that selected patients might benefit from using clinafloxacin. Such is the case of a diabetic cystic fibrosis patient suffering pulmonary infection with *Burkholderia* [17].

New non-fluorinated FQs have also been designed mostly during the last two decades. Nemonoxacin is one of them with activity against Gram-negative and Gram-positive bacteria. Nemonoxacin has proven to be effective against methicillin-resistant *Staphylococcus aureus* (MRSA) and quinolone resistant *C. difficile*. Importantly this new drug possesses antimicrobial activity against *Helicobacter pylori* and vancomycin resistant pathogens [18,19,20,21]. A detailed focus on the nemonoxacin development as a therapeutic agent is presented in a review article by Qin and Huang, 2014 [22].

Potentiation of quinolones antibacterial effects may help to keep quinolones effective. Although still at an experimental state, several studies propose the use of a variety of molecules to potentiate antimicrobial effects of quinolones, which can be divided into agents with and without antimicrobial effects themselves. Example of the former includes naturally occurring compounds such as ferulic acid, a phenolic phytochemical found in plant cell walls and a good example of compounds with antimicrobial activity that potentiates quinolones effects [23,24]. Some others are synthetic, including new generations of quantum dot compounds [25]. Many antibiotics have synergistic effects when combined with quinolones. Such synergistic effects are, in many cases, intended to broaden spectrum, rather than just simple increase activities [26]. Between agents without antimicrobial effects, that potentiate quinolones, vitamin K_3_ inhibits the function and expression of NorA efflux pump on *S. aureus*, where it actively expel the quinolone norfloxacin off the cellular envelop [27]. Recently, Lim et al. described a small molecule constructed by modifying pipemidic acid, (an early quinolone molecule with discrete antimicrobial activity) with the capacity to inhibit DNA-repair processes [28]. The small molecule, identified as IMP-1700, works the same way as quinolones, its planar structure allows for intercalating within the DNA sequence and binding of therapeutic target, the AddAB/RecBCD repair complex, the same way quinolones reach and inhibit gyrase and topoisomerase. Inhibition of mismatch repair system RecBCD was described to increase sensitivity to quinolones [29]. The small molecule IMP-1700 and future molecules based in IMP-1700 seem very promising as adjuvant or stand-alone therapies. A series of 3-phenylquinolone, constructed by modifying the isoflavone biochanin A, resulted in very effective efflux-pump inhibitors in *Mycobacterium avium*, reducing MIC of macrolides and FQs. These inhibitors represent the first specifically designed compounds exhibiting potent inhibitory activity of efflux pumps. A summary of agents increasing quinolones activity is presented in Table 1. Putative, and well-described targets at both the protein or the gene level have been described to modulate antibiotics resistance, including quinolones; using either target specific or global search approaches. Many genes and proteins affecting resistance are membrane factors or are related to metabolism. These topics are covered in the section about the mechanism of antimicrobial activity of quinolones and the mechanism of resistance.

The concern of drug-and multidrug-resistance (MDR) of bacteria has led beyond just adding new members with improved features to families of existing antimicrobials, including the quinolones family. The use of combined therapies, therapies that include coadjuvants and therapies that use conjugated antibiotics such as hybrid antibiotics has been proposed [23]. A study with a clinical strain of *Acinetobacter baumannii* MDR, susceptible to colistin, showed that the combination of colistin plus levofloxacin was synergistic and bactericidal in vitro. The results of the infection test using a non-mammalian model showed an increase in survival when the combination colistin-levofloxacin was used, compared to colistin alone [35]. Berry et al. described that polybasic peptides conjugated with levofloxacin, with the addition of an aliphatic group tether, resulted in compounds that potentiate ciprofloxacin, levofloxacin, and moxifloxacin against MDR isolates of *Pseudomonas aeruginosa*, *E. coli*, *K. pneumoniae* and *A. baumannii* [23]. The hybrid antibiotic tobramycin-ciprofloxacin recovers activity of FQs against MDR *P. aeruginosa* in vitro and enhances FQs efficacy in vivo [36]. The presence of tobramycin and ciprofloxacin, separated by a C12 tether, is necessary for function as adjuvant. Studies indicate that antibacterial mechanisms of ciprofloxacin are conserved, while tobramycin activities of the hybrid are limited to destabilization of the outer membrane. New studies by the same group indicate that tobramycin-moxifloxacin hybrid structure increases outer membrane permeability and decreases the efflux by reducing the proton motive force though the membrane in *P. aeruginosa*. The optimized drug prototypes present a significant effect by protecting *Galleria mellonella* (wax moth) larvae from lethal effects of MDR *P. aeruginosa*. In addition, the optimized hybrids reduced the level of resistant bacteria selected over a period of 25 days of exposition to the hybrid (twice the MIC), compared to treatments with moxifloxacin (16 times the MIC) or tobramycin (512 times the MIC). Although the hybrid possesses potent activity against MDR *P. aeruginosa* isolates, the activity can be synergized when used in combination with other antibiotics [37]. The structures and use as stand-alone antimicrobials or as an adjuvant is extensively discussed in “Antibiotic Hybrids: the Next Generation of Agents and Adjuvants against Gram-Negative Pathogens?” by Domalaon et al. [38].

## 3. Physicochemical Properties of Quinolones

The antibacterial activity of quinolones is influenced by the physicochemical properties. In this regard, one of the most relevant characteristics of these antibacterials is the presence of ionizable functional groups [39]. Most of the molecules belonging to this family are zwitterions, with an isoelectric point ranging from 6.8 to 7.8, therefore, at physiological pH the dominant occurring forms are zwitterions, followed by the neutral form [40,41]. Currently, FQs whose predominant species at physiological pH is not a zwitterion are under development and exist in clinical use. Delafloxacin, a fourth generation FQ registered for use in humans, is an anionic non-zwitterionic molecule, with poor presence of cationic microspecies, only at extreme acidic pH and uncharged at acid pH [42,43,44]. As the balance of the species is pH-dependent, the solubility of FQs also varies depending on the medium and the solvent in which it is found. Along the gastrointestinal tract, quinolones will encounter different physicochemical environments that largely differ from the site of infection. The neutral species has the lowest solubility in water and other polar solvents, whereas in a non-polar environment quinolones solubility is higher, facilitating its passage through lipid membranes. On the contrary, the zwitterionic species has a greater solubility in a polar environment, which would facilitate its passage through pores of bacterial membranes [45]. Therefore, the antibacterial activity and the bioavailability of fluoroquinolones is influenced by the physicochemical properties of the solvent, modifying the enthalpy and entropy in the formation of non-covalent bonds with the environment [45]. It has been analyzed that the negative values of the free energy of solvation of FQs are related to the spontaneity of the dissolution of the compound both in an aqueous medium and in the ease of crossing biological membranes. These data are directly related to the good bioavailability that FQs have in general [45,46].

Molecules of the FQs family complex with divalent and trivalent metallic cations, forming water-insoluble complexes [5]. The proximity of the oxygen atoms of the deprotonated carboxyl group with the oxygen atom of the carbonyl group favors the formation of coordinated bonds with Mg^2+^, such interaction largely depends on pH [47,48]. This physicochemical property has a direct impact on the bioavailability of FQs. FQs complexes with polyvalent cations including Cu^2+^, Fe^2+^, Ni^2+^, Zn^2+^, Al^+3^ would limit diffusion through the epithelial barrier and might result in subtherapeutic concentrations of these antimicrobials [49,50]. Thus, when administered orally, FQs require appropriate administration to avoid polyvalent FQ-metal interactions to ensure adequate absorption and effective systemic concentration [49].

Likewise, some FQs have also been found to act as competitive ligands for amino acids that complex with these metals in blood plasma. However, under normal conditions of administration, they do not have an impact on the mobilization of metals from plasma proteins in the bloodstream [51]. Studies of norfloxacin translocation kinetics through *E. coli* OmpF nanopores have shown that Mg^2+^ and Ca^2+^ decrease the residence time by increasing the permeability to norfloxacin through OmpF, this interaction is dependent on the concentration of divalent cations [52]. Another study that investigated the uptake of norfloxacin at different pH and concentrations of divalent metal ions, found that the presence of divalent metals causes a decrease in the incorporation and that acidic conditions also decrease the uptake process [53]. FQs cross the cell membranes of both Gram-positive and Gram-negative organisms via passive transport [54]. Studying the accumulation of pefloxacin performed in *S. aureus* has shown that FQs without a net charge penetrate the cytoplasmic membrane easier than charged species. These observations suggest that entry of FQs dependent on the dominant microspecies in the extracellular environment [55]. Likewise, studies have found that the neutral species is more lipophilic than the zwitterionic species [56]. Consistent with this, studies have shown that only the neutral species crosses the lipid membrane and the zwitterionic species remains in the aqueous phase. However, when the zwitterionic species approaches the membrane, it forms aggregates, which decrease polarity, favoring their translocation through the lipid bilayer [41].

The distribution of nalidixic acid and nine fluoroquinolones between 1-octanol and aqueous buffers has been studied at 25 °C, pH 5, 7 or 9 and an ionic strength of 0.15. The partition coefficients were determined by analysis of both the organic and aqueous phases using either liquid chromatography or spectrophotometry. The partitioning of the compounds studied was generally consistent with that of zwitterions so that the highest partitioning was observed when the pH of the aqueous solution was close to that of the isoelectric point (pH 7) and decreased at both higher (9) and lower (5) pH values. However, deviation from ideal partitioning behavior was observed at pH 5 that was attributed to ion-pairing between the cationic form of the solute and anionic buffer species. Relationships between the theoretical partition coefficients obtained by a functional group approach and the experimental values obtained here were perturbed by the complex ionic equilibria of the compounds [56]. 

The chelating properties of quinolones have been explored for the development of metallo-antibiotics. Ternary complexes FQ-Cu(II)-phenanthroline have shown similar antimicrobial activity against *E. coli* and greater antimicrobial activity against MRSA compared with conventional FQ formulations [57,58]. The translocation pathway through the bacterial cell of the FQ-Cu(II)-phenanthroline complexes would be different and could be a new strategy for the development of molecules, to face bacterial resistance [58].

## 4. Synthetic and Natural-Occurring Quinolones

Although quinolones are synthetic antimicrobials, their occurrence as a natural product is well documented in plants, but also bacteria. One case is the compounds present in *Tetradium ruticarpum* (synonymous *Evodia rutaecarpa*). Fruits of this plant are used in Asia for dispelling cold, soothing liver, and analgesia. Further research demonstrated that alkaloids are part of active ingredients of *T. ruticarpum* and demonstrated the presence of evocarpine, a natural quinolone active against *H. pylori*, an enterobacteria causative agent of peptic ulcers and gastric cancer [59,60]. These studies also describe several other related quinolones with the potential as antimicrobial. All of them need to be evaluated as possible therapeutic agents. Furthermore, some 4-quinolones isolated from the conditioned culture broth of the actinomycete *Pseudonocardia spp*. decreased growth of *H. pylori*. The compound with the higher antimicrobial potency was the one with the epoxide group (Figure 2, fourth compound), which showed a minimal inhibitory concentration (MIC) of 10 mg/L and an even more pronounced bacteriostatic effect [60,61]. These naturally occurring quinolones present a geranyl or oxidized geranyl side chain at position C-2 compared to fatty acid-derived alkyl or alkenyl chains usually found in microbial quinolones. Although these compounds have been described over twenty years ago, no therapy has been developed based on these quinolones; however, some advances have been made in the synthesis of those structures [62]. We will be watchful for the development of new therapies based on the structures of naturally occurring quinolones. 

## 5. Structure-Activity Relationship (SAR) of Quinolones

The pharmacophore of the FQs family is constituted by a system of two fused rings, which form the derivatives of the naphthyridone core (for example, nalidixic acid and trovafloxacin; see Figure 3 for details of the chemical formulae, as well as for the rest of molecules mentioned in this chapter) and the derivatives of the quinolone core (most of molecules in the quinolones family; Figure 1) [4,63,64]. In both groups, the structure of ring A (1,4-dihydro-4-oxo-3 pyridine carboxilic acid) is conserved. The nucleus B varies and can be pyridine or another aromatic or heteroaromatic ring. The presence of a nitrogen in position 8 identifies the series of naphthyridones and a carbon, in position 8, to the series of the quinolones [63,64]. The minimum chemical structure required for antibacterial activity is the presence of a non-reducible double bond at position 2–3 and a free ketonic group at position 4 [63,65,66].

The introduction of the basic piperazinyl group at position 7 of nalidixic acid led to the design of pipemidic acid, the first piperazinyl quinolone. Pipemidic acid showed improved activity against Gram-negative microorganisms [67,68]. Almost parallel to this discovery, flumequine, the first fluorinated quinolone, was synthesized. In this molecule, the hydrogen at position 6 of the core quinolone was replaced by a fluorine atom, increasing the potency with respect to the non-fluorinated quinolones. This molecule increased antimicrobial activity by 2, 4 and 15-times against *K. pneumoniae*, *E. coli* and *Proteus vulgaris*, respectively, compared with nalidixic acid. In addition, flumequine also showed an 8-times increased activity against *S. aureus* [69]. This led to designing molecules that combined the piperazinyl substituent in position 7 and a fluorine atom in position 6 leading to the first fluoroquinolone, norfloxacin, that showed an increase in activity against Gram-negative and Gram-positive bacteria and an improvement in pharmacokinetic, with respect to nalidixic acid [67].

The change of the ethyl group in the position N1 of norfloxacin with a cyclopropyl group in ciprofloxacin improved antimicrobial activity 4 to 32-times, with respect to norfloxacin against different microorganisms. These improvements translate, for example, into greater antipseudomonal activity and broadening the spectrum against Gram-positive and anaerobes [70,71]. The antipseudomonal effects are sensitive to piperazine nitrogen atom substitution, whose change produces loss of antipseudomonal activity [63]. The combination of fluorine at C6, the piperazinyl ring at C7, and cyclopropyl at N1 increased the potency of ciprofloxacin to inhibit the activity of *E.coli* gyrase nearly 200-times, relative to nalidixic acid [66,72]. Importantly, the introduction of fluoride explains the improvement in the pharmacokinetic profile of FQs and became the dogmatic characteristic of the group, however, today there are new non-fluorinated molecules such as nemonoxacin [5,44,73].

Most FQs are bicyclic derivatives, however, flumequine combines a tricycle from which molecules such as ofloxacin and its stereoisomer levofloxacin emerged [5]. In this group, a third ring of 6 atoms is formed between nitrogen 1 and carbon 8 of the main core. This third ring produces restriction in rotation and the introduction of a chiral center, in the methyl linked to oxazine, which resulted in an increase in the spectrum of activity against Gram-positive [5,74].

The combination of a methoxy group at C8, a diazabicycle in position 7 and a cyclopropyl in position 1 gave rise to moxifloxacin, a fluoroquinolone with activity against Gram-positive, atypical bacteria, *Mycobacterium tuberculosis* and anaerobes [64,75]. The methoxy group at C8 would be related to its increased activity against anaerobes [64,66]. Molecules with methoxy group substituents at C8 or halogens have been reported to have higher activity against resistant strains of *S. aureus* and *Mycobacterium smegmatis* compared to ciprofloxacin [76]. Today, approved molecules with a Cl^−^ at position C8, such as besifloxacin, which also replaces the piperazinil group with a 3-aminohexahydro-1H-azepine, expand the global activity against Gram-positive [44]. Therefore, C8-halogenated derivatives with a fluorine, such as lomefloxacin, have been associated with a higher incidence of photosensitivity [5,66]. 

Various FQs have been withdrawn from the market due to severe safety concerns, including those with an N1-difluorobenzene such as temafloxacin and trovafloxacin that caused severe hemolysis and hepatotoxicity, respectively [5]. Sparfloxacin and grepafloxacin, from the family of C5 substituents, were also removed for cardiotoxicity [5,44]. 

Since their discovery, thousands of molecules have been synthesized and despite the safety problems presented by some fluoroquinolones, the study of new substituents and the introduction of new fluoroquinolones in the pharmaceutical market has continued [44,77]. Examples of the above are the FDA approvals of the FQs besifloxacin, delafloxacin, and finafloxacin. For an in-depth review of their characteristics and their role in therapeutics, we suggest the review by Rusu et al., here they describe introduced modifications that might shape the fifth generations of quinolones [44].

The FQs have been used for almost 50 years and are today fundamental for antimicrobial chemotherapy. Adverse reactions of some quinolones such as arthropathy, increased QT interval, and effects on the nervous system has been described [66]. Drug–drug interactions via CYP-450, conditions quinolones use in selected groups of patients. In addition, age, pathologies, and physiological conditions like pregnancy must be considered. Therefore, further research on the relationships of the structure that might maintain and/or improve antimicrobial potency with an adequate safety profile should continue to produce improved molecules [5,44,78].

SAR studies have been instrumental to develop new quinolones drugs that constitute the four generations quinolones and the molecules that will conform the fifth generation. With the advances in quantitative structure-activity relationship (QSAR), new drugs with improved potency and new capacities regarding spectrum have been developed. QSAR studies in combination with molecular docking and molecular dynamics simulation will be important, not only to improve potency and to modulate spectrum, but also to manipulate structural motifs to produce quinolone analogues that lack undesired effects. FQs interact with type II topoisomerases through a H_2_O-Mg^2+^ bridge. Computational studies were performed to understand these interactions and will be helpful to design molecules that minimize off target binding via H_2_O-Mg^2+^ bridge [79]. QSAR modeling has been used to design new quinolone analogues and conjugated drugs. Quinolones-FQs conjugates linked by amino acids were generated. Using a QSAR multilinear model, antimicrobial activity was predicted and resulted close to experimental MIC, with conjugates of oxolinic acid and ciprofloxacin as the most potent molecules against *Salmonella* Typhi, *Staphylococcus aureus* and *Streptococcus pyogenes* [80]. Drug interactions with plasma proteins have recently been studied to develop a reliable drug design through computational predictions. For this, the knowledge about quinolone molecules in combination with computer-aided drug design (CADD) and SAR were used. The study found fluoroquinolones derivatives with lower binding to plasma proteins. This is important as fluoroquinolones bound to plasma proteins act as strong antigens, inducing the production of specific antibodies against the drug that indirectly affect white blood cells by binding to drug-plasma proteins bound to leucocytes, producing leucopenia [81]. Reduced plasma protein binding would also increase distribution of quinolones through tissues. These authors also used the predictive models of bioconcentrations and photodegradation of trovafloxacin derivatives [82,83]. They found that 11 trovafloxacin derivatives decreased bioconcentrations by 36.90–61.41% and presented photodegradability by 9.04–20.56%, compared to trovafloxacin [81]. Therefore, the evidence suggests that computer aided simulations and design facilitate and decrease costs of drug designs [84].

## 6. Other Biological Activities

Throughout this work we have mainly talked about the use of quinolones as antimicrobials; however, various uses have been described that currently represent an alternative in diverse therapies due to their broad mechanism of action. As the quinolones keep homology and set its origin during the attempts to synthesize the anti-malarial agent chloroquine, it is logical to explore the potential of quinolones as antiparasitic agents. However, the efficacy of the FQ norfloxacin, as anti-malarial, was accidentally discovered during treatment of patients suffering *Salmonella* Typhi infection. Later, its efficacy was tested in patients suffering falciparum malaria [85,86,87]. A randomized study demonstrated that chloroquine cleared plasmodium and its symptoms faster than norfloxacin. Since that, most of the FQs approved for use in humans have shown important anti-plasmodial activity [88,89]. However, it has been reported that *Plasmodium falciparum* develop resistance to specific FQs and other treatments. Therefore, newly synthesized molecules based on quinolones and the related structures of quinolines are under investigation for their potential as anti-parasitic drugs [90]. One example is the 4-quinolone-3- carbohydrazide that has been used as a platform for designing new compounds, with the potential to be used as anti-plasmodial agents [91]. Since the late 1980s studies have shown anti-trypanosomal activity, first for quinolones such as nalidixic and oxolonic acid derivatives [87]. Later this activity has been described for multiple FQs molecules. Additionally, anti-trypanosomal activity of FQs was increased by producing Cu(II)-complexes [92]. More recently, the related molecules, quinolone-amides were used to design anti-trypanosomal compounds with many of them presenting in vivo activity [93].

At first, tested quinolones did not present anti-fungal activity. Therefore, a search of quinolones derived molecules with anti-fungal activity concentrated in synthesis of new molecules, hybrid molecules, and metal complexed quinolones. In addition to broad spectrum and strong antibacterial effects, several clinafloxacin-triazole hybrids presented anti-fungal activity against *Candida albicans* and *Candida mycoderma* that was stronger compared to the fluconazole control [94]. Lomefloxacin complexed with Cr^3+^, Fe^3+^, Cu^2+^ and other metallic ions showed higher activity against *C. albicans* than the free ligands [95]. Furthermore, the FQ gatifloxacin complexed with metals increased its anti-fungal activity toward *Tricophyton rubum*, *C. albicans* and *Fusarium solani*. While the complex of nalidixic acid with Ag(I) metal ion showed increased antifungal activity compared to nalidixic acid against four fungi: *Pythium aphanidermatum*, *Sclerotinia rolfsii*, *Rhizoctonia solani* and *Rhizoctonia bataticola* [96,97]. Recently, new 4-hydroxy-2-quinolone-analogs were described with at least one of the new molecules presenting promising anti-fungal activity toward *Aspergillus flavus* [98].

The idea of inhibiting viral helicases as a therapeutic target comes from the observation that prokaryotic helicase can be inhibited by molecules, such as quinolones, to reduce or stop DNA replication. Helicases are essential proteins present in eukaryotic organisms, bacteria and viruses and are indispensable and necessary to resolve DNA and RNA double-stranded structures, consequently, specific sequences of single-stranded DNA or RNA are available for processes such as replication and transcription [99]. Early observations showed that inhibitors, such as quinolones, inhibits BK-polyomavirus replication and its cytopathic effects [100,101]. Inhibition of the Simian virus 40 large T antigen helicase activity by levofloxacin, ciprofloxacin, ofloxacin or trovafloxacin inhibits viral DNA replication and SV40 virion replication, measured as plaque formation in an in vitro assay [102]. 

Inhibition of viral replication by quinolones has been observed by different mechanisms in vaccinia virus, herpes simplex virus, influenza A and HCV [99,103]. Many FQs inhibit the propagation of HCV and replication of HCV genome by inhibiting viral helicase [103]. The potential of current quinolones and new derived molecules to inhibit viral replication is promising and need to be further explored. The challenge of exploring these new therapeutic applications should include SARS-CoV viruses which encode their own helicase [104]. In addition, an in silico study suggests the potential of quinolones to interact and probably to inhibit SARS-Cov-2 protease [105].

Immunomodulatory effects of quinolones have been extensively reported mostly in in vitro models and are mainly anti-inflammatory, however, their mechanisms as immunomodulators are not fully understood [106,107,108,109]. Recently, ciprofloxacin and levofloxacin were reported to inhibit the microglia inflammatory response mediated by NF-kB by inhibiting signaling of LPS through toll-like receptor 4 (TLR4) [110,111]. In human peripheral blood monocytes, moxifloxacin inhibits secretion of IL-8, IL-1beta and TNF-alpha secretion whose production was induced by infection with *Aspergillus fumigatus*. The evidence indicates that moxifloxacin inactivates MAP-kinases ERK1/2, p38 and the p65-NF-κB signaling pathway [112]. The FQ moxifloxacin, inhibits the proinflammatory effects of the anti-cancer drug etoposide [113]. Despite the evidence revealing anti-inflammatory effects, there are no clinical recommendations of quinolones as anti-inflammatory drugs.

The effects and possible use of quinolones as anti-cancer drugs have been reported alone or in combination with other drugs. The focus has been on developing new quinolone drugs, naturally occurring quinolones, hybrid quinolone-based molecules and quinolone-metal complexes [114]. Induction of cell cycle arrest and apoptosis have been reported for FQs, such as ciprofloxacin, in several cells used for the study of cancer models [115,116,117,118]. Complexes of moxifloxacin and copper presented anti-proliferative and apoptosis-inducing effects in several breast cancer cell lines [119]. Along with moxifloxacin inhibition of the pro-inflammatory effects of anti-cancer drug etoposide, it enhanced etoposide antiproliferative and apoptotic effects in THP-1 and Jurkat cells [113]. Lomefloxacin, that could cause phototoxicity, produced apoptosis in HL-60 human promyelocytic leukemia cells, when irradiated with ultraviolet A (UVA) light, opening a venue in the study of quinolones as light-induced drugs with potential for spatiotemporal therapies. Reduction in apoptotic cells and caspase-3 activation was observed in presence of the quencher histidine, suggesting the importance of photodynamically-generated singlet oxygen as mediator of apoptosis [120]. Although the discovery and engineering of new quinolones intended as anti-cancer agents, is of great interest, fulfilling specificity for malignant tissue, without poisoning normal tissue, will be challenging. Extensive revision of the anti-cancer effects of FQs has been covered previously [118,121].

Other effects of quinolones, such as inhibition of cytochrome P-450 family-enzymes are listed in Table 2. Effects of quinolones-related molecules are also included in the table.

## 7. Clinical Importance of Quinolones

After nalidixic acid, the first member of quinolones used for treating urinary tract infections, FQs have had better potency, absorption, and tissue distribution for additional uses in treating urinary tract infections, sexually transmitted diseases, digestive tract infections, respiratory tract, skin and infections of bones and joints [4]. The clinical tolerability of these agents at usual doses has been generally acceptable [131].

The quinolones available for clinical use have been classified into four generations, mainly on the basis of their spectrum of activity, therefore, inclusion of specific molecules in each generation may differ between reports [132]. After the molecule flumequine, the second generation of quinolones had the main characteristic of a fluorine atom at position 6, collectively known as fluoroquinolones, which presents a marked increase of antibiotic activity [133]. These compounds were most potent against Gram-negative bacteria, therefore, its activity against *Streptococcus pneumoniae* was minimal to justify a clear indication for its use, to treat respiratory tract infections. Soon after, the emergence of resistance reduced its potential against *S. aureus* [132]. 

Quinolones of second generation, ciprofloxacin and ofloxacin are widely used today, with ciprofloxacin being the most active against *P. aeruginosa*. Interestingly, for the chiral molecule ofloxacin only the *S*(−) isomer presents antibiotic activity [134]. This isomer has been marketed as levofloxacin, which is twice more active than ofloxacin, without major changes in its action spectrum [135]. On the other hand, sparfloxacin and grepafloxacin, members of the third generation, must be considered separately since their substituent at position 5 and 7 significantly enhanced their antibiotic activity against *S*. *pneumoniae*. However, both agents were quickly withdrawn for toxicological reasons [136]. Sparfloxacin presented marked phototoxicity which manifested as erythema, while dysgeusia is between the most important secondary effects of grepafloxacin [136]. A further improvement in activity against Gram-positive bacteria, along with significant anaerobic activity, was observed within the fourth-generation molecules, most likely caused by the presence of an alkyl-substituted (piperazine or pyrrolidine) at position 7, and of a methoxy at position 8 [137]. In this way, trovafloxacin, was one of the most active compounds and had the broadest spectrum, but it was soon restricted to treating serious infections in the US, and withdrawn in Europe, due to hepatotoxicity effects [138]. In summary, clinical data support that ciprofloxacin is the most active compound against Gram-negative organisms, that moxifloxacin is preferentially active against Gram-positive organisms, that ofloxacin and levofloxacin show intermediate activity, and that norfloxacin is an intrinsically weak fluoroquinolone against Gram-positive organisms [135].

In cases of urinary tract infections, quinolones are not recommended as first line treatment for acute uncomplicated infections. Instead, nitrofurantoin possesses antibacterial effects compared with FQs, but with less effects over gut-microbiota [139]. In addition, seven days of combined trimethoprim/sulfamethoxazole has shown to be as effective as seven days of ciprofloxacin, for the treatment of pyelonephritis [140]. Among ambulatory patients treated with fluoroquinolone for UTIs, there was evidence of superiority of norfloxacin or ofloxacin compared with ciprofloxacin or Trimethoprim/sulfamethoxazole in terms of treatment failure [141].

On the other hand, remarkably ciprofloxacin, ofloxacin, and the newer FQs reach significant intracellular concentrations. Therefore, moxifloxacin, gatifloxacin, levofloxacin, and the investigational drug gemifloxacin have excellent activity against *Legionella*, *Chlamydia*, *Mycoplasma*, and *Ureaplasma* species [142]. Intracellular respiratory pathogens such as *Mycoplasma pneumoniae*, *Chlamydia pneumoniae*, and *Legionella pneumophila* are predictably susceptible to FQs [142]. These antibiotics are regarded as second-line antituberculosis agents and should be reserved for the treatment of resistant *M. tuberculosis* [131]. Although children are an important target population with respect to infections that respond well to quinolones, such as diarrhea or Gram-negative meningitis, the combined risks of toxicity and the rapid spread of resistance should contraindicate treating children with quinolones, with the possible exception of children with cystic fibrosis or life-threatening infections with organisms resistant to other antibiotics [131]. 

Another such use for quinolones, as a second line therapy, is the case of *Campylobacter spp.* persistent infections by resistant strains. In these cases, using macrolides, FQs (e.g., ciprofloxacin), and tetracyclines is recommended. Note that no antimicrobial therapy is recommended to treat self-limited minor infections by *Campylobacter* [143]. In the treatment of *H. pylori*, FQs were first considered as a promising first line therapy, before rapid appearance of resistance. Today FQs, such as levofloxacin or moxifloxacin, are recommended in triple therapy in combination with amoxicillin. Perhaps, FQs can be considered first line treatment for *H. pylori* in places were prevalence of resistance to FQs remains low and resistant to clarithromycin is high [144].

## 8. Mechanism of Antimicrobial Activity of Quinolones

Quinolones and FQs are inhibitors of nucleic acid synthesis [145,146]. Its antimicrobial activity is derived mainly by interfering with the activity of type IIA bacterial topoisomerases, DNA gyrase and topoisomerase IV [147]. Type II topoisomerases catalyze transient cleavage (Gate (G) -segment), the transport of intact double-stranded DNA through transient cleavage (Transfer (T) -segment) associated with the use of ATP and resealing of the broken strands [147,148]. Gyrase relieves stress that originates in front of the replication fork by removing positive supercoiling and is the only type II topoisomerase that can introduce negative supercoiling into the genome. It is found in bacteria, plants and Archaea and was identified as the first target of quinolones in 1977 [148,149,150]. Topoisomerase IV has been described mainly in bacteria and is responsible for the decatenation of chromosomes behind the DNA replication fork, removing intermolecular cross-linking of DNA [148]. 

These enzymes participate in the maintenance of the DNA topology and, therefore, in maintaining the homeostasis of the supercoiling state, being essential for cell growth and survival [147,148]. Changes in the supercoiling balance have been shown to alter gene expression [6]. For example, in *S* Typhimurium changes in DNA topology mediated by gyrase have been correlated with changes in the expression of genes involved in adaptation to the environment found in the macrophage [151].

Both enzymes, DNA gyrase and topoisomerase IV, are tetramers made up of heterodimers of homologous GyrA2/GyrB2 and ParC2/ParE2 subunits, in Gram-negative species respectively [147,152]. The homologous subunits of topoisomerase IV in Gram-positive species are GrlA2 and GrlB2 [152]. Studies based on analysis with *E. coli* strains that had mutations in gyrase, topoisomerase IV, or both enzymes, concluded that gyrase is the main target for quinolones and that topoisomerase IV is the secondary target [153,154,155]. Genetic evidence found in other Gram-negative microorganisms, such as *H. pylori*, *P. aeruginosa*, *Serratia marcensces*, and *Campylobacter jejuni* revealed DNA gyrase as the main target [156]. Consistent with these conclusions, quinolones are more potent against *E. coli* gyrase than against *E. coli* topoisomerase IV [152,154]. Later, studies in *S. pneumoniae* found that topoisomerase IV was the main target for ciprofloxacin [157,158]. This led to the concept that gyrase is the main target for quinolones in Gram-negative bacteria and topoisomerase IV is the main target for quinolones in Gram-positive bacteria [152]. Although this idea is widely accepted, the evidence indicates that the main target varies within each microorganism and with each FQ [159,160,161]. Delafloxacin, a FQ recently approved for clinical use, has shown affinity for DNA gyrase and topoisomerase IV, from both *E. coli* and *S. aureus* [42,161,162]. In *S. pneumoniae* and *S. aureus* trials with blesifloxacin, an FQ for use in ophthalmic infections, showed that it similarly inhibited both gyrase and topoisomerase IV. Studies with other FQs such as moxifloxacin, gatifloxacin, and gemifloxacin, indicate that the main target in *S. aureus* is topoisomerase IV [163]. The selective toxicity of the FQs is based on the higher affinity for prokaryotic topoisomerases compared to eukaryotic topoisomerases [72]. However, despite higher selectivity, they are not free of clinically relevant adverse effects [164]

To evaluate antimicrobial treatment with FQs, the pharmacokinetic/pharmacodynamic (PK/PD) parameter that best predicts its microbiological activity and clinical efficacy is the ratio f-AUC 24/MIC, which is generally >30 for Gram-positive microorganisms, >125 for Gram-negative microorganism and ~ 50 for anaerobes, with variations between individual microorganisms. FQs show a concentration-dependent pattern of bactericidal activity for Gram-negative microorganisms, whereas for Gram-positive and anaerobic pathogens it shows a time-dependent pattern of bactericidal activity [165].

As can be seen in Figure 1, FQs have different substituents, however C2 (hydrogen atom), C3 (carboxyl group) and C4 (carbonyl group) are fundamental for their antimicrobial activity, and they remain unchanged in the different molecules of the family [63,65,66,166,167]. For the formation of the ternary complex DNA-quinolone-gyrase, the presence of Mg^2+^ chelated by FQs via the oxygen atoms of the carbonyl and carboxylic group is essential. It has been suggested that this complex would establish coordination bonds with 4 water molecules, which in turn, through hydrogen bonding, establish interactions with DNA and with an aspartic/glutamic and a serine of the helix IV of the GyrA subunit of the gyrase enzyme [166,168]. Analysis of the crystal structures of the DNA-enzyme-quinolone complexes reveal that the position N1 and C8 groups intercalate in the nicks of DNA introduced by topoisomerases, and substituents such as the fluorine atom and the heterocycle group bind to the enzyme [166,169]. The substituent group at position C7 of quinolones, orientates toward GyrB/ParE subunits of gyrase and Topo IV respectively [168,169,170]. An alternative orientation has been suggested as a possible binding site to form the FQ-enzyme-DNA complex, in this model the piperazinyl group at C7 interacts with GyrA [168,170]. 

Quinolones bind noncovalently to gyrase or topoisomerase IV that are bound to DNA, apparently before DNA cleavage occurs, causing a conformational change that induces slow DNA cleavage and prevents gyrase and topoisomerase IV mediated DNA religation [171,172,173]. Thus, the binding of quinolones would be involved in two steps, one that occurs before DNA cleavage and another that occurs after cutting. [174]. In this last step, the FQs binding site is exposed once the DNA-bound enzyme has cleaved a 4-base pair-separated segment of double-stranded, intercalating between cuts and stabilizing the cleavage complexes [147,148,152,166,175,176,177].

Death mediated by FQs has been described as a two-step mechanism. First, the formation of reversible DNA-enzyme complexes stabilized by quinolones and second, the subsequent collision with the replication fork, causing fork arrest without releasing cut DNA from double stranded DNA, blocking DNA replication and bacterial growth as a consequence [167,168,174,178]. These effects on DNA, would be related to the bacteriostatic effects of these antimicrobials and would be insufficient to cause quinolone-mediated cell death [167,174,179]. Second, after the formation of these reversible complexes, two pathways have been described that would occur downstream and that would explain the bacterial-killing mediated by these antimicrobials. An independent pathway of protein synthesis and aerobic growth and another pathway dependent in protein synthesis and anthaerobiosis have been described. Chloramphenicol and anaerobic shock have been found to interfere with this pathway, by blocking quinolone-mediated chromosome fragmentation. The first-generation quinolones lost their lethality under these conditions, however, norfloxacin was insensitive to anaerobic conditions, but did not show lethality in the presence of protein synthesis inhibition. Also, ciprofloxacin maintained its lethality in the presence of protein-synthesis inhibition and required higher concentrations to achieve lethality in anaerobiosis [174,180]. The events required for the generation of double-strand break are not fully understood, it is thought that for the quinolone-mediated death pathway independent of protein synthesis, these drugs would destabilize the quinolone-gyrase-DNA complexes by separating the subunits of the gyrase which would lead to a release of fragmented DNA [147,174].

It has been postulated that chromosome fragmentation would lead to an accumulation of ROS and that ROS would contribute to the lethality observed in some quinolones. Studies indicate that the bactericidal effect of oxolinic acid was blocked when chloramphenicol or a combination of thiourea plus 2,2′-bipyridyl were used, which inhibits the production and accumulation of hydroxyl radicals, suggesting that these events are in the same pathway [167]. However, with these inhibitors, DNA fragmentation was not blocked, as it was observed with chloramphenicol, suggesting that ROS accumulation occurred after chromosome fragmentation and is essential in stress-mediated death [167,181]. The discovery of a general ROS-dependent mechanism that contributes to the killing process of bactericidal has been controversial, regarding its true role in the bactericidal mechanism [181,182,183].

In this regard, new evidence indicates that quinolones and other bactericidal and non-bactericidal antimicrobials stimulate auto accumulation of ROS, which is essential for cell death [181,184]. Hong et al., carried out tests where *E. coli* was exposed to nalidixic acid, then nalidixic acid was removed before seeding *E. coli*, in agar plates with or without the antioxidant agent bipyridyl. The presence of bipyridyl increased bacterial survival around 30 times, at the same times that reduced accumulation of ROS. While in the absence of bipyridyl accumulation of ROS was observed. These authors provide evidence that nalidixic acid-mediated toxicity promotes accumulation of ROS that continues even after the antimicrobial has been removed, and that in turn was responsible for the observed lethality [181]. The same effect was observed in conditional mutants that were subjected to lethal stress associated with ROS accumulation, the addition of thiourea or catalase, prevented ROS-mediated death [181]. Finally, these authors propose that some antibacterials, including quinolones, cause primary damage, which if severe or within a repair-systems deficient bacterium can kill them directly, without the need for ROS-dependent mechanisms. This primary damage would activate the pathways for ROS accumulation, which would lead to secondary damage and more ROS accumulation, overcoming the repair pathways causing loss of viability and death [181]. In experiments that evaluated the paradoxical survival observed in bacteria when high concentrations of quinolones are used, it was found that the concentration of nalidixic acid, that caused the highest bacterial death, correlates with the maximum detection of ROS. At higher concentrations of quinolones, the presence of ROS decreased, which would contribute to the observed tolerance of *E. coli* to high concentrations of quinolones [184]. These findings provide data on the key role of ROS in quinolone-mediated death and help in the understanding of the effects of quinolones after their interaction with their main drug receptors. At the same time, this observation opens a window for the identification of new antimicrobial targets for future antimicrobial therapies [183].

## 9. Mechanisms of Resistance to Quinolones

It has been described that the selection of resistance genotypes depends on the concentration of FQs. At low concentrations, mutants unrelated to the target are selected, at moderately high concentrations mutants are selected on the target, and at higher concentrations mutants unrelated to the main target are selected again [167]. The resistance observed in clinical strains is given by a gradual accumulation of mutations that decrease the intracellular concentration or decrease the affinity for the target. There are mechanisms of resistance to quinolones encoded in chromosome and in plasmids. The former, frequently corresponds to mutations in the genes that encode gyrase and topoisomerase IV, resulting in decreased binding of quinolone with the enzyme, therefore, losing its ability to inhibit DNA ligation or to form stable ternary complexes [181,183]. It has been found in using an standard *E. coli* strain that the evolution of *gyrA* mutations confer high resistance and have an impact on bacterial fitness, as seen for other mutations that confer resistance [185]. In *S.* Typhi, mutations in *gyrA* were found to increase fitness [186]. However, in clinical strains of *M. tuberculosis*, susceptible to FQs, it was found that genetic variations modulate the evolution of resistance to FQs through *gyrA*, affecting the fitness and MIC of different strains [187].

The cellular concentration of quinolones is regulated by the opposite actions of entry by diffusion and exit mediated by efflux pumps, as well as enzymatic degradation [152,188]. In contrast to Gram-positive species, the outer membrane of Gram-negative bacteria creates an additional barrier that the drugs must cross to enter the cell. Hydrophobic FQs can partially diffuse through the lipid bilayer, but most FQs, including ciprofloxacin, are hydrophilic and their ability to cross the membrane is limited. Porins, especially OmpF, become the main pathway for the translocation of FQs [189]. Studies that evaluated the interaction of ciprofloxacin with membrane models in Gram-negative bacteria show that the entry of ciprofloxacin through the bacterial membrane depends on OmpF [189,190]. It has been described in different publications that the loss of OmpF and OmpC porins is related to resistance to antimicrobials, including quinolones, especially for *E. coli*, *S.* Typhimurium and *S*. Typhi, this mechanism is encoded in the chromosome [152,164,191,192,193,194,195,196].

The FQs and other antimicrobials are actively expelled from the bacterial cell through pumps anchored to the inner membrane that are ubiquitously distributed in all domains of life. To date, there are 6 families, ABC, MFS, SMR, MATE, PACE and RND. Many of the pumps described confer a phenotype of resistance to more than one antimicrobial [197]. Among the pumps described that confer resistance to FQs in clinically relevant Gram-negative microorganisms are found, for example, the AdeABC efflux system, AbeS in *A. baumannii*, the AcrAB-TolC system, OqxAB-TolC, EmrAB-TolC from *Enterobacteriaceae*; EfpA, Mmr and Rv1258c from *M. tuberculosis* and MexAB-OprM, MexCD-OprJ, MexEF-OprN in *P. aeruginosa*. Efflux pumps are also frequently found in Gram-positive bacteria such as NorA in *S. aureus* and PmrA and PatAB in *S. pneumoniae* [197]. Recently, a new efflux pump encoded by the *abaQ* gene that confers resistance to FQs has been described in *A. baumannii* [198].

Efflux-mediated resistance to FQs occurs by constitutive expression of transporters or overexpression of genes involved in transport and is controlled by global and specific regulators [167,199]. Efflux-mediated resistance contributes to the intrinsic and acquired resistance, while its regulation is complex and highly interconnected with the physiological state of the bacteria. In addition, efflux might participate in the bacterial adaptation to environmental stress. The mechanisms of resistance regulate gene expression to increase resistance by orchestrating a reduced influx through the repression of porins and increasing the output of antimicrobials through efflux [197]. Among the global regulators that confer resistance to FQs are the Mar, Ram, Rob, as well as two-component systems such as SoxRS [200,201]. Participation of local repressors like AcrR and RamR is also part of the final phenotypes of resistance that regulate expression of efflux pumps. Participation of sRNAs as final regulators or proteins that participate in the efflux or influx of quinolones is also extensively described. The resistant phenotype does not fully depend on the balance of influx/efflux. For example, it has been described for the *marRAB* (multiple antibiotic resistance) operon that in addition to modulating the expression of pumps and porins, it is capable of activating *xseA* expression, which codes for an exonuclease VII involved in DNA repair, reducing damage induced by FQs [202]. Regarding SoxRS, experiments with strains of *E. coli* resistant to FQs with mutations in *gyrA*, showed an increase in transcription of *soxS*, due to a mutation that leads to the production of truncated SoxR that constitutively activates *soxS*, which resulted in an 8-times increase of *sox* expression and twice the expression of *marA*. These changes result in a further increase of the AcrAB efflux system and repression of expression of porins, which in turn may result in increased MIC for norfloxacin and ciprofloxacin by 64- and 16-times [200]. The regulatory proteins SoxS, MarA, Rob and MarA recognize genes that contain the regulatory sequence “marbox” in their promoters and have overlapping regulatory effects, however, they bind to this box with specificity and differential affinity [202].

During the last years, there are several classic and OMICs experimental reports relating metabolism with resistance to antibiotics. Mechanistically speaking, it has been described that changes in the metabolic state participate in the activation of master regulators such as *marA* and *ramA* and consequently in the phenotype of resistance to antibiotics, including FQs. In *S*. Typhimurium, defective in *cysJIH* genes, that participates in cysteine biosynthesis, the addition of ciprofloxacin activated the transcription of genes encoding the regulators *marA* and *ramA*, as well as the genes related to the efflux system *acrB*, *tolC*, *smvA* and *ompD*, probably inducing efflux of toxic compounds. In addition, *cysteine*-auxotrophy decreases the accumulation of reactive oxygen species (ROS). Apparently, affecting the biosynthetic pathways of cysteine increases resistance to ciprofloxacin, but also to the oxidative agent paraquat though both, increasing efflux and detoxifying enzymes [203]. In *S*. Typhi, it was recently reported that inactivation of the glutamine-synthetase gene *glnA* produces a glutamine auxotrophic strain, along with susceptibility to the FQs, such as, ciprofloxacin. The authors found that *glnA* inactivation in *S.* Typhi promotes the expression of porin OmpF, a well described entry channel for quinolones, at both transcriptional and protein level. The Increase in OmpF may be explained by repression of small RNA MicF expression and increased transcription of *glnL-glnG* two-component system. MicF downregulates *ompF* transcripts, while the two component-system NtrB/C (encoded by the genes *glnLG*) positively regulates OmpF [204]. The two-component system CbrA/B encodes for a sensor and a response regulator that controls metabolism, including the fate of nitrogen and carbon in *P. aeruginosa*. This two-component system also controls synthesis of LPS. It has been reported that mutants in the genes encoding CbrA/B are sensitive to ciprofloxacin, but also polymyxin [205,206]. 

There is intricate crosstalk between resistance to quinolones and bacterial metabolism at genomic and non-genomic levels. Figure 4 shows a model that integrates metabolism and resistance to quinolones. Many metabolic changes can also affect the resistance to antimicrobials different from the quinolone family. The presence, intake and metabolization of carbon sources, such as sugars and amino acids, increases the proton motive force and the efficient activity of efflux pumps that may expel antimicrobials, including quinolones [206,207]. Genomic effects of metabolites include affecting expression of genes involved in enzymatic reactions, metabolites uptake and outer membrane proteins to facilitate the intake of small molecules. Presence of certain metabolites, including sugars different from glucose, may increase expression of inner membrane permeases, competing for space in the inner membrane, affecting efflux of antimicrobials. This last effect has been demonstrated for antimicrobials such as tetracycline, chloramphenicol, meropenem and several biocides in bacteria such as *Salmonella*, *P. aeruginosa* and *A. baumannii* [208,209], the effects on quinolones resistance need to be explored.

Besides the direct effects of quinolones, inhibiting replication of nucleic acids, it has been described that treatment with ciprofloxacin increases expression of genes involved in the multiresistance phenotype of bacteria, detoxifying enzymes, and efflux pump systems [203]. In all cases, the final phenotype of resistance to quinolones and other antibiotics will be the contribution of genomic and non-genomic effects due to the presence of small metabolites and antibiotic molecules.

After more than two decades of the discovery of plasmid-mediated resistance to quinolones (PMQR), the genes of resistance to quinolones that can be mobilized have been described mainly in Gram-negative bacteria of clinical importance, such as *E. coli*, *P*. *vulgaris*, *Shigella*, *Salmonella*, *K. pneumoniae*, *Enterobacter*, *A. baumannii*, and *P. aeruginosa* [210,211,212,213,214,215,216,217,218]. To date, three resistance mechanisms have been identified in transferable elements [77,219]. The first mechanism is mediated by Qnr proteins that protect gyrase or topoisomerase IV from inhibition by quinolones. These proteins are encoded by the *qnr* genes of several families with multiple alleles, including the *qnrA*, *qnrB*, *qnrC*, *qnrD*, *qnrE*, *qnrS*, and *qnrVC* genes [77,210,211,212,213,214,215,216,217,218,220]. Qnr proteins are members of the PRP family (from pentapeptide repeat protein) that contain domains composed of repeating pentapeptides sequences of approximately 200 amino acids and it has been proposed that they act as DNA mimetics competing with DNA for its binding to type IV gyrase or topoisomerase [221]. The three-dimensional structure of the first member of the PRP family was determined for the fluoroquinolone resistance protein MfpA from *M. tuberculosis*, this protein binds to DNA gyrase and inhibits quinolones activity [219].

Qnr proteins appear to confer resistance to quinolones by two different mechanisms. Like MfpA, they decrease the binding of gyrase and topoisomerase IV to DNA, protecting bacteria from the action of quinolones by reducing the number of enzymes available on the chromosome [219]. Also, these proteins would bind to the enzyme competing with quinolones and FQs [222]. Recently, it has been shown that the presence of nalidixic acid, norfloxacin or ciprofloxacin QnrB19 from *Salmonella* was able to restore supercoiling activity, in a concentration-dependent manner, reaching a maximum protective activity and then inhibiting supercoiling. A possible explanation would be related to its mechanism of action by competing with the DNA gyrase substrate, such as relaxed or positively supercoiled DNA, due to its structure that somehow may mimic double-stranded DNA [216]. It was also shown that QnrB19 increased the IC50 of ciprofloxacin and norfloxacin but not nalidixic acid, suggesting that these differences are related to structural differences that would affect the potency of quinolones in the presence of QnrB19 [216]. QnrB19 is the predominant allele in *Salmonella* (>80%) and has been found in plasmid Ppab19-4, transduction has been proposed as a possible transfer mechanism, for propagation of the plasmid [223].

The origin of the *qnrA* genes has been identified in *Shewanella*, for the *qnrB* genes it has been suggested in *Citrobacter*. Other *qnr* genes have been found on the chromosomes of *Vibrionaceae*, all microorganisms from aquatic environments [77]. Plasmid-encoded quinolone resistance genes have been described as causing a low level of resistance but may facilitate the selection of chromosomal resistance mechanisms, conferring high levels of resistance, as a consequence [167]. It was recently shown that Qnr is a DnaA-binding protein that activates DNA replication stress, leading to increased mutation rate. These effects would be independent of their interaction with DNA gyrase and provide a mechanistic basis for explaining how the presence of these proteins facilitates selection of mutants with a high level of resistance to FQs [224,225].

The second mechanism confers resistance by enzymatic modification of a variant of an aminoglycoside transferase that acetylates the unsubstituted nitrogen of the piperazine ring that has ciprofloxacin and norfloxacin in its structure, thus reducing the activity of these antimicrobials, as well as, that of some aminoglycosides and is encoded by the *aac(6′)-Ib-cr* gene [77,226]. It is a narrow spectrum mechanism inactivating only two FQs, however, it is widely distributed in *Enterobacteriaceae* [210,211,212,213,214,215,216,217,218,219]. Recently, in *P. aeruginosa* an ATP-phosphotransferase encoded by the *crpP* gene that inactivates ciprofloxacin has been described, finding homologous genes in clinical strains of extended-spectrum β-lactamase (ESBL)-producing enterobacteria, suggesting that this new gene may be highly distributed in Enterobacteriaceae [227]. An *aac(6′)-lb-cr* gene with identical sequences was found in clinical strains of *E. coli* and strains of *Sporosarcina*, *Rhodococcus*, *Kytococcus*, *Erythrobacter* from marine sediments impacted by aquaculture, indicating that flow of genes between these bacteria has probably been recent [226]. 

The third group of plasmid-encoded resistance proteins is comprised of efflux pumps, with OqxAB, QepA1 and QepA2 being identified, up to date. QepA (quinolone efflux pump) protein belongs to the MFS family, it is capable of eliminating hydrophilic FQs (ciprofloxacin and norfloxacin) by actively expelling the drugs, they were first identified in strains of *E. coli* of hospital origin in Japan and France, respectively [228,229]. OqxAB, an expulsion pump of the RND family, was initially identified in transferable plasmids responsible for resistance to olaquindox, used for inducing growth in pigs [230,231]. OqxAB has broad substrate specificity, including quinolones such as ciprofloxacin, flumequine, norfloxacin, nalidixic acid, and antibacterials, such as chloramphenicol and trimethoprim [232]. The prevalence and distribution of plasmid-encoded quinolone resistance genes varies between countries, environments, and microorganisms. A recent study, in aquatic environments of China, found a high prevalence of *qepA* together with *aac(6′)-Ib-cr* with respect to *qnrB* [233]. In contrast, in Latin America, the prevalence of *aac(6′)-Ib-cr* and *qnrB* was the most abundant in samples of diverse origin [218].

In an effort, to find drug targets that act as adjuvants that prolong the useful life of these antimicrobials, different investigations show strategies to enhance or restore the efficacy of FQs. Recacha et al., deleted *recA* in genetic constructs of *E. coli* with chromosomal and transferable mechanisms of resistance. The *recA* deletion caused a decrease in MIC for different FQs tested. The authors found that suppression of the SOS response, as a consequence of *recA* deletion, increased the bactericidal activity of ciprofloxacin and reduced survival in resistant strains, providing evidence that suppression of the SOS pathway can reverse FQs resistance [234]. Recently, sRNA screening identified candidate genes whose inactivation restored susceptibility to ciprofloxacin in strains with a *gyrA* S83L substitution that confers a high level of resistance in *E. coli* [235]. On the other hand, investigations with palmatine, an isoquinolone alkaloid, extracted from plants, showed resensitization of *E. coli* strains with plasmids harboring *qnrS* and *aac(6′)-lb-cr*. The authors found that this compound is able to bind to functional sites of both *qnrS* and *aac(6′)-lb-cr* and has synergy with ciprofloxacin, both in vitro and in vivo, proposing it as a candidate with potential for inhibition of PMQR [33]. These results open alternatives for finding new molecules and new targets to increase susceptibility or reverse resistance to FQs.

## 10. Veterinary Use of Quinolones and Animal Production

This family of broad-spectrum antibacterials has been used in areas different than human medicine. These include, uses in both terrestrial and aquatic food animals, as well as in the clinic to treat diseases in large and small pet and sport animals [236]. Quinolones were introduced for use in food animals in the late 1980s and early 1990s [237]. Within terrestrial livestock species, quinolones and FQs are used in considerable amounts, in poultry farms, pigs, cattle. While in aquatic species the use of quinolones extends to salmonids and other species of fish, as well as in shrimp [236,238,239,240,241,242].

In cattle, the fluoroquinolone danofloxacin is licensed for the treatment of respiratory diseases associated with *Mannheimia haemolytica* and *Pasteurella multocida* [243]. Another extensively used quinolone is enrofloxacin, licensed for use in cattle, for the treatment of respiratory disease associated with *M. haemolytica*, *P. multocida*, and *Histophilus somni*. In pigs enrofloxacin is used for the treatment of respiratory diseases associated with *Actinobacillus pleuropneumoniae*, *P. multocida*, *Haemophilus parasuis*, and *Streptococcus suis* and for the treatment and control of colibacillosis, caused by pathogenic *E. coli* strains [243,244]. Various FQs such as ciprofloxacin, difloxacin, flumequine, norfloxacin and quinolones such as oxolinic acid are registered for use in poultry [245]. Therapies based in these drugs have shown useful in the treatment of colibacillosis and other infections caused by *E. coli* in poultry, as well as in the treatment of avian mycoplasmosis [246]. 

In China, the FQs ofloxacin, ciprofloxacin, levofloxacin, norfloxacin, and gatifloxacin have been used to treat colibacillosis in poultry [247,248]. In the United States, the off-label use of the FQs is prohibited in animal production, due to the potential of antibiotics to select for resistant bacterial strains that may threaten human health. Danofloxacin and enrofloxacin are the only FQs authorized for use in sheep, cattle and pigs. While their use has not been approved in poultry since 2005 [243,247]. The ban on the use of FQs in poultry was based on the increase in the incidence of infections produced by *Campylobacter spp*. resistant to FQs in humans, associated with the consumption of poultry [249]. However, in countries such as the United Kingdom, Poland, China and Brazil their use is authorized in poultry and other food animals [238,250]. A similar situation occurs with the use of quinolones in aquaculture, its use is authorized in salmonid producing countries such as Norway and Chile, while in the USA, the use of FQs is not authorized in aquaculture [251].

The amounts of quinolones and FQs used in animal production differs between countries and types of animals. Today there is no precise estimate of their global use, in animal production. Data on the quantities used are obtained from active surveillance systems, mainly in developed countries, with little or no information from countries with low and middle income. Lack of systematized track of information makes difficult to determine the quantities and potential risks of FQs for human and animal health. This lack of information is especially critical when it comes to those antimicrobials classified as critically important. Keeping traceability will be critical to establish policies aimed to reduce antibiotics consumption and use, as a global response to the threat of resistance [252,253]. Today the World Organization for Animal Health (OIE) has developed procedures to collect data from member countries regarding antimicrobials used in animals, in response to the WHO global action plan to contain antimicrobial resistance [254,255]. In its third report, it indicates that between 2015 and 2017 the quinolones and FQs corresponded to 2.7% of all antimicrobials used in animals. In terrestrial, aquatic animal production and in pet animals, their use corresponded to 2.9, 2.5, and 5%, respectively. Despite providing an overall view of antimicrobial use by species and by geographic zones, the report does not provide details regarding use by specific country and still has a few years with information coverage to really appreciate changes in use trends [255].

Regarding quantities of quinolones used in the United States, 24.5 Tons of FQs, representing 0.4% of sales of antimicrobials for veterinary use, were used during 2019 [256]. According to the sales of antimicrobials for animal production, reported by the FDA, between 2013 and 2019 the percentage of FQs marketed for antimicrobials used in livestock production has remained below 1%. However, the percentage of variation in this period shows an increase in FQs use by 62.6%, with sales amount of 15.1 Tons in 2013 that increased to 24.5 Tons in 2019 [256,257,258,259,260,261,262]. 

In Europe, the trend in the use of quinolones and FQs in animal production has decreased by 7.6% between 2011 and 2018, selling 192.5 Tons in 2011 and 178 Tons in 2018. During this same period, this family of chemotherapeutics represented on average 2.5% of antimicrobials used in animal production in Europe [263,264,265,266,267,268,269,270]. In Denmark, an average of 1.05 Tons of oxolinic acid were used in marine corps, between 2014 and 2016. In Norway, the use of quinolones in farmed fish, with Atlantic salmon representing more than 95% of the biomass of farmed fish, the use of oxolinic acid decreased from 926 kg in 2009 to only 66 kg in 2019 [271]. Regarding the use of quinolone and FQ in land-based production animals, it has not exceeded 20 kg per year between 1993 and 2018 [272]. The use FQs in 2019 in cattle pigs and sheep corresponded to 0.1, 0.04, and 0.01% of total used, respectively [271]. Data on sales of antimicrobials for veterinary use in France indicate that the percentage of FQs corresponded to 0.21% and quinolones to 0.58% of the total during 2018, representing a low percentage of the total antibiotics used, with FQs exposition decreased by 86.1% compared to 2013. In this country, the variation in sales of FQs and quinolones between 2011–2018 in mg of active ingredient per kilogram of live weight was −80.2% and −53.9%, respectively [273].

Data on the use of quinolones and FQs in Latin America are scarce [238,274,275,276]. However, some studies have been carried out in countries with relevant levels of animal production. In Chile between 1998 and 2008, an average of 44 Tons of oxolinic acid were imported per year for aquaculture use [242,277,278]. Based on the precautionary principles, the Chilean aquaculture industry drastically decreased the use of quinolones since 2009, importing only one Ton between 2009 and 2015. Official sources of the State of Chile indicate that an average of 13 Tons of FQs have been used per year in animal production, with an average use of almost 10 Tons per year in terrestrial species and an average per year of 3 Tons in aquaculture, between 2015 and 2018 [279,280,281,282,283]. The use of FQs in Chilean aquaculture corresponded mainly to flumequine and according to research, between 1998 and 2015, 643 Tons of this antimicrobial were used in aquaculture in the southern part of the country for salmonid farming [242,252,277,278,284,285].

China is the largest meat producer in the world, leading production along with Europe and North America [286]. Based on antimicrobial sales data from different sources, some observations can be made regarding the use of antimicrobials and particularly FQs. Data from 2013 indicate that the total amounts of antimicrobials for animal production in the USA and Europe are comparable, while China exceeded the amount of antimicrobials used by almost 10 times compared to the USA and Europe in 2013. In addition, the percentage of fluoroquinolone use also differs. In China, it was found that the fluoroquinolone family represented almost 30% of antimicrobials used for animal production with 22,210 Tons, while they represented 2.3% and 0.2% in Europe and USA respectively, see Table 3. China used in 2013, 118 and 1480 times more FQs than Europe and the USA, respectively. The main use of FQs in China was to produce pigs with 13,900 tons, being ciprofloxacin (3110 Tons), enrofloxacin (3090 Tons) and norfloxacin (2820 Tons) the most used FQs in this species [239]. It is accepted that the consumption of antimicrobials is the main force for the selection of resistant microorganisms [6,287,288]. Thus resistant bacteria and resistance genes could move within and between different environmental compartments [289]. The use of FQs in animal production can be a source of resistance genes that represent a risk to human health [290]. In this regard, an investigation found that there is a positive correlation between the use of FQs in animal production and resistance to these antimicrobials in indicator microorganisms such as *E.coli*, *Salmonella* and *Campylobacter*, its results suggest that small increases in the use of these antimicrobials increase resistance [291]. Another study that evaluated resistance in microorganisms isolated from foods of different origins, found phenotypic resistance to FQs in strains of *Salmonella* Enteritidis, *Campylobacter* and *C. difficile* [292]. Another report that examined the presence of known determinants of resistance to quinolones, found in *Salmonella* Kentucky strains of human and animal origin with high resistance to ciprofloxacin and nalidixic acid, a double mutation in *gyrA* at codon 83 and 87 and a double mutation in the *parC* gene at codon 57 and 80, suggesting a clonal spread of the strain in the human and animal population of the studied area [293].

## 11. Impact of Quinolones in the Environment

As drugs intended to eliminate bacteria, the release of antibiotics into the environment has the potentials to affect bacterial communities in different ways including: alteration of phylogenetic composition of bacterial communities, spread of antibiotics resistance and perturbance of function of the bacterial communities in ecosystems [294]. Among the variety of compounds, directly or indirectly, verted to the environments, antibiotics rise the attention mainly because of the amount that have been used and the impact in the development and spread of resistance and multiresistant microbes. First intended to cure and prevent diseases in humans and animals, antibiotics have been found as micro-contaminant of soil and water, but also the tissue of animals and plants living in contaminated ecosystems [294,295,296,297].

The FQs present chemical, physicochemical, pharmacokinetic (metabolic and excretory), pharmacological and microbiological characteristics that have been studied and together account for their impact as environmental pollutants. From a chemical point of view, the FQs used in humans and in veterinary medicine are antibacterial of synthetic origin. Most FQs have at least two ionizable groups that are relevant to the typical pH values encountered in the environment. Common examples of ionizable groups are the 3-carboxyl group and protonatable group at C7-position, such as the 7-piperazinyl heterocycle of ciprofloxacin, norfloxacin, and enrofloxacin (see Figure 5 for details of ciprofloxacin microspecies) [298]. The FQs contain acidic or basic groups, the different dissociation constants of their substituents indicate that they have different charge depending on the pH of the medium. The pK_a_ of the carboxyl group has an approximate value of 6.0 ± 0.3, which is independent of the substituent in position 7 [65,299,300]. The pK_a_ of the basic function at position 7 is, on average, 8.8 [65]. FQs could exist in four forms of pH-dependent protonation [301]. In this regard, studies have shown that the zwitterionic and neutral species are the predominant forms at physiological pH [301]. For example, at pH 7.04, the isoelectric point of ciprofloxacin, coexist two predominant ionized states of the molecule [65,302]. Regarding the *n*-octanol/water partition coefficient, three groups of molecules can be distinguished: those such as ciprofloxacin, norfloxacin and enrofloxacin with a coefficient lower than 0.1, considered as hydrophilic FQs, those derivatives classified as weakly hydrophilic such as ofloxacin, levofloxacin and moxifloxacin with a coefficient between 0.1 and 2 and the quinolones nalidixic acid, oxolinic acid and difloxacin considered as hydrophobic derivatives with a partition coefficient greater than 2 [65]. This coefficient is one of the parameters to evaluate the differential distribution characteristics of different chemical products in environmental matrices, it has been used to identify, for example, the surprising affinity and distribution in solids of different active pharmaceuticals principles [302]. According to the logK_ow_ of −1.74 at pH 7.04 for ciprofloxacin that classifies it as a hydrophilic molecule, a smaller proportion would be expected in mud or sediment compared to that found in water [302]. Despite reported logK_ow_, ciprofloxacin, a hydrophilic molecule, and other FQs have been found to sorb well in mud, sediment and soil [298,302]. It is thought that ionic interactions of acidic and basic groups of quinolones have a role in this behavior, in a pH-dependent manner [300,302]. The solubility, hydrophobicity/hydrophilicity and the distribution are also pH-dependent physicochemical characteristics of ciprofloxacin [302].

Among other characteristics of FQs, it was found that their antibacterial activity is also affected by physicochemical factors such as pH due to changes in the ionization of the molecule, as well as the presence of divalent cations such as Mg^2+^ and Ca^2+^, which form complexes governed by ion-dipole interactions using 4-keto oxygen and the ionized 3-carboxylic acid group [39,303]. These variations in pH and environments rich in divalent metals can be found in different environmental matrices [65,300]. According to values of Henry’s Law constants at room temperature, FQs are classified as non-volatile compounds and have a high tendency to sorption in solids, including soil [299,300]. Likewise, different reports indicate that the characteristics of the soil (pH, salinity and temperature) influence the adsorption of FQs in sediments [299,300,304]. Huang et al., determined the concentrations of FQs from different environmental sources, finding that detected FQs concentrations represent a risk for human health and the environment [305].

The quinolones are not different in the effects that other antimicrobials have once they enter the environment [305]. They produce short term effects directly affecting microbial communities by modulating fitness, eliminating some microbials populations and modulating bacterial metabolism in different ways. Quinolones will also produce long-term indirect effects, such as the selection of resistant mutant microbes that can resist in different ways the presence of the antimicrobials.

In humans and animals, FQs are mainly eliminated through kidneys without changes. However, some quinolones such as moxifloxacin, grepafloxacin, and trovafloxacin are eliminated primarily by biliary excretion [65,164,306]. In the liver, FQs are metabolized to different degrees including, glucuronidation, *N*-oxidation and demethylation reactions that have been described to a variable degree. Fundamentally, two sites in the core structure of quinolones are subject to metabolic changes: the aromatic nucleus attached to position 7 and the carboxyl group at position 3, the latter fundamentally undergoing glucuronidation [65]. Results of metabolism indicate that FQs undergo poor metabolism, which may translate into high chemical stability [65]. For example, a study of difloxacin metabolism and excretion kinetics in pigs found that after oral administration, more than 95% of the antibacterial was excreted unchanged, and only a minor part was metabolized primarily to sarafloxacin, another fluoroquinolone [307]. For enrofloxacin, an FQ used only in veterinary medicine, a study of excretion kinetics found that after oral administration 74% was excreted unchanged and 25% was metabolized to ciprofloxacin, the FQ most used in humans [308]. Among the factors that affect the stability of FQs, forced degradation studies and simulated solar irradiation studies have shown that photolytic conditions can promote FQs degradation, this being the main degradation pathway on aquatic surfaces [300,309,310]. Alkalinity conditions have also been reported to promote the degradation of FQs [309].

Considering the physicochemical characteristics and the characteristics of their metabolism described above, it is plausible to think of FQs as chemical entities that remain and exert biological effects once they have entered the environment, after excretion or biotransformation in humans and/or animals that have been treated with these antibacterials. Detection of FQs in different environments has been reported [304,311]. From human sources, different FQs have been detected in urban wastewater, hospital wastewater, waste from pharmaceutical plants, and wastewater treatment plants [300,312].

Studies of river water and water from hospital effluent have identified FQs such as ciprofloxacin, norfloxacin, ofloxacin [313,314]. Of the FQs identified in river water, their origin was determined to come from human sources [313]. Likewise, quinolones have been identified in manure, wastewater, veterinary hospital effluent, sediment, mud, and fresh and saltwater bodies from veterinary sources [300,315]. In a study that identified different antimicrobials in effluent water from a veterinary hospital, FQs reached the highest concentration among all identified antibacterial [316].

FQs produce short term effects directly affecting microbials communities by modulating fitness and/or eliminating some microbial populations and modulating bacterial metabolism in different ways [185,186,317,318]. Quinolones will also produce long-term indirect effects, such as selection of resistant mutant microbes that can resist in different ways the presence of antimicrobials and other ecotoxicological threats, by mechanisms that are poorly understood [319,320].

Literature indicates that FQs enter the environment in considerable amounts and have been considered a micropollutant that is difficult to remove [300]. When released in its active form, it can persist and spread in the environment, affecting exposed communities. Exposure to FQs residues (depending on the concentrations) can result in antimicrobial activity, altering the balance of the indigenous communities and the selection of resistant microorganisms with the consequent potential for transmission of resistance genes by horizontal gene transfer (HGT) [295,321,322]. 

Conte et al., found ciprofloxacin residues and microorganisms resistant to this FQ in water samples from sanitary and hospital effluents, treated water from a wastewater treatment plant and from river surface water. Among the determinants of resistance, microorganisms with single point mutations and double mutations in the GyrA QRDR were identified. The presence of PMQR was also found, identifying the genes *aac-(6′)-lb-cr*, *oqxAB*, *qnrS* and *qnrB* [323].

A study revealed increased fitness in strains transformed with plasmids containing *qnr* genes, derived from *K. pneumoniae* obtained from wastewater, in the presence of sublethal concentrations of ciprofloxacin, compared with strains without the plasmid. This change was not dependent on the size or copy number of the plasmids studied, but rather on increased *qnr* transcription, suggesting that the expression of these genes is regulated by the concentration of ciprofloxacin. The concentrations tested in this study were like those found in sewage treatment plants, showing that the presence of *qnr* confers a selective advantage in presence of these antimicrobials [324].

A study carried out in India to study contamination with quinolones from discharges of a pharmaceutical industrial zone, intended to identify the presence of *qnr* genes in samples of well water, soil, and river sediments found it at 42%, 7% and 100% of the samples, respectively. The same authors compared the frequency of *qnr* genes in river sediment in Sweden, which was 18% [322]. Likewise, fecal samples with at least one *qnr* gene were higher in individuals from India than in individuals from Sweden, 91.6% vs. 24.3%. In the same vein, Tomova et al., have shown that quinolone-resistant uropathogenic *E. coli* strains isolated from patients in a coastal region with intensive aquaculture in Chile harbored significantly more *qnrB* and *aac-(6′)-lb* genes and less *qnrA* compared to strains from New York patients [325].

The differences observed between India and Sweden and between Chile and the United States, could be related to differences in regulation, regarding the use and prescription of FQs in human medicine, with more restrictions for the use of quinolones in countries such as the United States and Sweden [322]. However, the studies carried out by Tomova et al., also found that strains of marine bacteria, from areas impacted by using antimicrobials in aquaculture, and strains of uropathogenic *E. coli* share identical *qnr* genes. In addition to the *qnr* genes, structural similarity in class 1 integrons, identical gene cassettes and sequence similarity in their flanking regions were found, suggesting that use of antimicrobials in aquaculture could facilitate HGT among these bacterial populations of different environmental compartments [326,327]. Figure 6 illustrates the mechanisms by which DNA might be acquired horizontally. 

The HGT is a natural process that actively contributes to the evolution and speciation of microorganisms. As illustrated in Figure 7 and Figure 8, the selection pressure exerted by antimicrobials in the ecosystems makes the prevalence of genetic determinants of resistance higher and propitiates the spread of resistance through HGT. An iconic case of multiresistance is *A. baumannii*, a Gram-negative bacterium with the capacity of up-taking foreign DNA from the environment that can be incorporated into its genetic patrimony. This microorganism, first described as an environmental bacterium, becomes today a multiresistant threat, with the case of some strains resistant to nearly all approved antimicrobial therapies [329]. Quinolones are no stranger to this problem and show a rapid decrease in effectiveness due to the rise of resistance. A study from Malaysia demonstrated that between 2003 and 2008, *Acinetobacter spp*. resistant to ciprofloxacin increased from 20% to 50% of clinically isolated country-wise [330,331].

Today the removal of antimicrobials, including FQs, represents a challenge for wastewater treatment plants. FQs clearance has been found to vary between 50% to 80% [321]. However, the fraction that is removed by conventional methods is frequently accumulated via adsorption in the mud (sludge), which is often used as fertilizer, representing a re-entry route with potential effects in the terrestrial environment, since FQs remain biologically active in the sludge [298,311,319]. In the search for solutions to eliminate FQs, the ability of some prokaryotic fungi and microalgae to biotransform and biodegrade FQs has been studied [311].

The ability of some fungi to metabolize FQs has been attributed to the low specificity of the substrate of their enzymatic battery, among which are the intracellular enzymes monooxygenases of cytochrome P-450 family and extracellular enzymes such as manganese peroxidase and hydrogen peroxide producing enzymes, among others [311]. Studies have shown the biodegradation of molecules such as flumequine, ciprofloxacin, enrofloxacin, ofloxacin and sarafloxacin, by a wide variety of fungi, including *Cunninghamella elegans*, *Pestalotiopsis guepini*, *Phanerochaete chrysosporium*. A major problem observed in studies, addressing this approach, is the residual antimicrobial activity after the biodegradation process, probably because some of the FQs metabolites also have antimicrobial activity. It is necessary to improve the identification of microorganisms capable not only of removing the parental drug, but also the metabolites, until ideally reaching the mineralization of FQs [332].

It is necessary to increase the current knowledge of contamination with quinolones in different environments because of anthropogenic activities, in that sense, there is a lack of more in-depth studies of the long-term impact that these antimicrobials cause in different ecosystems and how these changes affect human and animal health, but also plants and microorganisms. In an environment apparently highly polluted by FQs, it is also necessary to establish effective strategies to limit and reduce the amount that enters the ecosystems, applying the One Health concept and understanding that FQs are antimicrobials with the highest priority for human health [333].

## 12. Considerations for the Use of Quinolones: Human and Veterinary Medicine

Quinolones and FQs are effective drugs to treat multiple types of infections in humans and are effective against a wide variety of pathogens. For this reason, the WHO categorizes this family of antibacterials as critically important with the highest priority for their protection, since they are used for the treatment of bacterial diseases that affect humans [333]. However, today this family of antibacterial is also used in areas other than human medicine, which includes use in both terrestrial and aquatic food animals, as well as in the veterinary clinic of large and small animals. Within the main terrestrial food-producing uses of quinolones are poultry, pigs and cattle [238,275]. While, in aquatic species, their use is extended to salmonids (trout and salmons), tilapia, and shrimp [334]. It is expected that regulations on the use of quinolones reduce their use in animal farms. Alternatively, the development of prophylactic immunization also reduces the use of antimicrobials, such as quinolones, in animals and helps preserve their use for human medicine.

## 13. Recent Finding about Quinolones

Residual enrofloxacin and flumequine were found in clams from clam farms in Taiwan. Enrofloxacin is banned for humans and some farm animals (for example, poultry farms) for its toxicity. The authors conclude that although concentrations are below any toxic effect in humans, the presence of these agents has to be screened for safety issues [335]. Zarkan et al., found that inhibition of indole production increases the activity of quinolones against persisters of *E. coli* [336], probably by binding to the ATP binding site of GyrB subunit of DNA gyrase. The use of anti-virulence factors have been proposed as an antimicrobial therapy alternative to antibiotics [337]. Recently, compounds with anti-virulence activity, such as gallium (a siderophore quencher) and furanone C-30 (a quorum sensing inhibitor) in combination with ciprofloxacin and other antibiotics were effective in clearing infection and decreasing the spreading of resistant *P. aeruginosa* [338].

## 14. Future of Quinolones, Final remarks, and Conclusions 

There are no doubts about the importance of quinolones in human health. Therefore, it is important to save the efficacy of this class of drugs for several reasons. Quinolones are today, truly saving-life drugs covering a wide range of diseases and therefore are between the most prescribed drugs worldwide. However, the extensive use of quinolones in animal farms, but also in human medicine, increased prevalence of resistance in a rising number of pathogens. Therefore, it is a priority to restrict the use of quinolones to the use in human medicine only. In that regard, a global strategy is needed to ban the use of the same families of antibiotics in animal production and humans. Since the first quinolone antibiotics were introduced in the clinic, thousands of molecules in the quinolones family have been synthetized. The first designed drugs were more into the trial and error field, however, pioneer quinolone molecules allowed to learn the bases of the structure-activity relationship. This has resulted in the design of quinolones with increased antibacterial potency, spectrum, and penetration through tissues. The continuous learning about SAR combined with the advances in developing computing models to predict desired features of drugs will shape the coming generations of quinolones. The newly produced quinolones should focus on reducing undesired features including molecules that reduce off target interactions, reduction of drug-drug interactions. On the other side, it is urgent to design new molecules that overcome resistance to quinolones already in use.

## Figures and Tables

**Figure 2 molecules-26-07153-f002:**
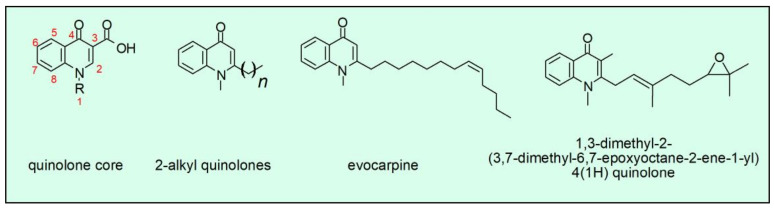
Comparison of the quinolone core, leading to the synthetic quinolone family of antimicrobials, with some of the naturally occurring quinolones, found in the plant *Evodia rutaecarpa* (the second and third chemical formulae) and a quinolone produced by the bacteria *Pseudonocardia* spp. (the fourth chemical formula, with the epoxy group). Chemical formulae were prepared using a free version of ChemSketch (ACD/Labs, Toronto, ON, Canada).

**Figure 3 molecules-26-07153-f003:**
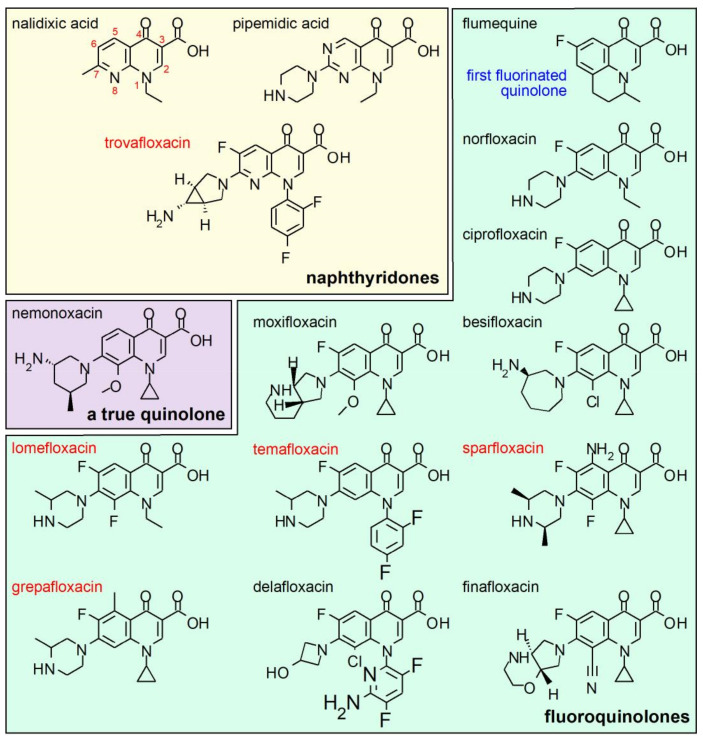
Features and functional groups found in the quinolones family of drugs. Colored polygons enclose chemical subclasses within the quinolones family of drugs. Red texts are indicative of withdrawn drugs due to safety concerns. Molecules are displayed according to mentioning within the text and do not necessarily appear according to time of discovery or marketed date. Chemical formulae were prepared using a free version of ChemSketch (ACD/Labs, Toronto, ON, Canada).

**Figure 4 molecules-26-07153-f004:**
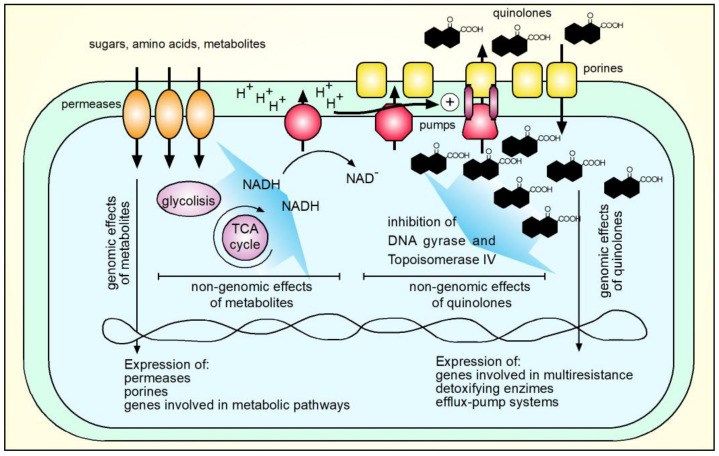
Integrative model of crosstalk between metabolism and resistance to quinolones. Metabolites, such as sugars and amino acids energize antibiotics efflux pumps through increasing proton motive force. In a different manner, these metabolites produce genomic effects, such as increasing inner membrane permeases, porins and metabolic pathway components that in turn may modulate the resistance to quinolones. Besides antimicrobial effects and expression of genes involved in quinolones resistance itself, quinolones may increase the expression of genes involved in resistance to other biocides. Figures were drawn from scratch using Freehands 10 (Macromedia, San Francisco, CA, USA).

**Figure 5 molecules-26-07153-f005:**
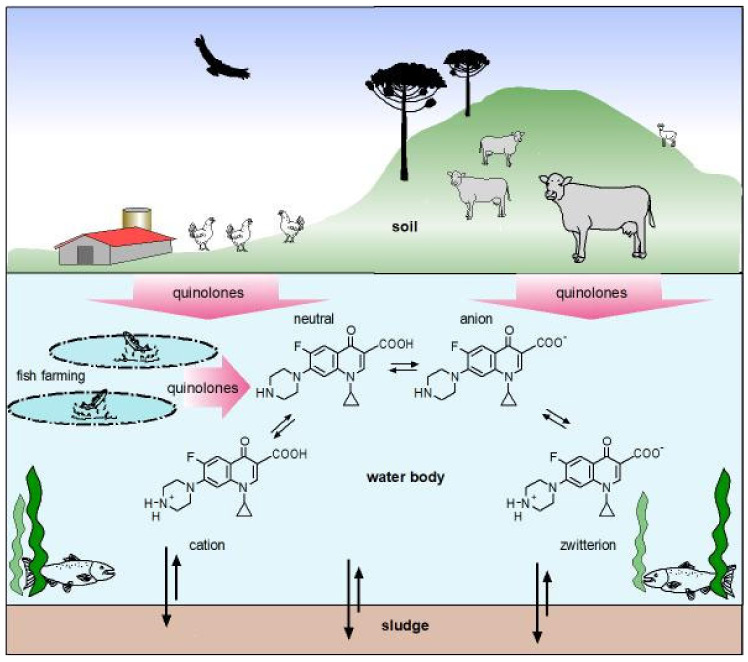
The quinolones and FQs, such as ciprofloxacin contain in their structure ionizable groups. They are responsible, at least in part, for their good distribution through tissues, but also solubilizing in the different environmental compartments. Many QFs have been found in sludges, water, and fisheries with the consequent effects in bacterial communities. Figures were drawn from scratch using Freehands 10 (Macromedia, San Francisco, CA, USA).

**Figure 6 molecules-26-07153-f006:**
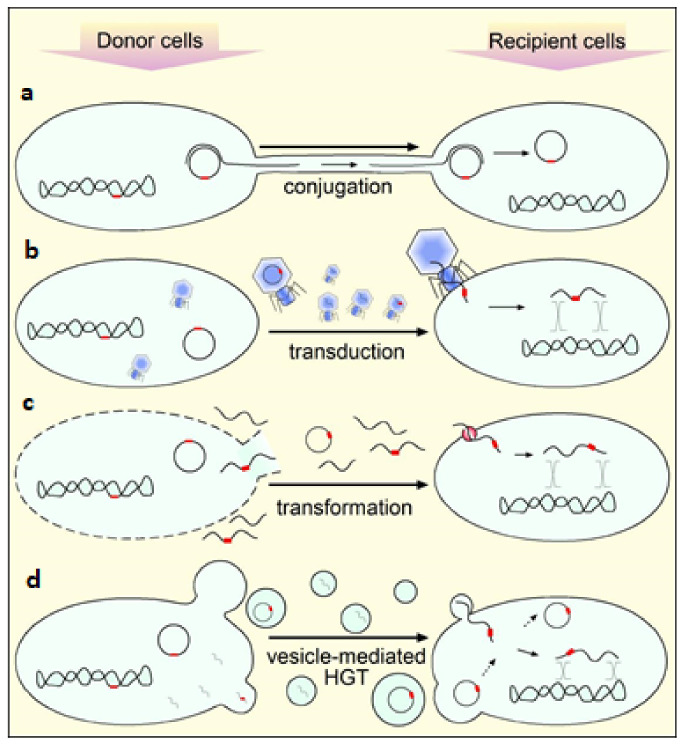
Genetic determinants of antibiotic resistance, including resistance to quinolones, can be horizontally transferred by four mechanisms. (**a**) Conjugation-mediated HGT depend in the presence of a fertility plasmid (plasmid F) in the donor bacteria, plasmid F encodes the functions to form and mobilize F plasmid through a sexual pili. In some cases, chromosomal DNA can be also transferred. (**b**) Transduction is the transfer of DNA mediated by bacteriophages, viruses that infect bacteria. Occasionally, the viral capsid may encapsulate bacterial DNA and transduce it into a recipient bacterium. Transduced DNA can be incorporated by homologous recombination or other mechanisms. In addition, viral DNA can be integrated to start forming part of bacterial chromosome. (**c**) Transformation is the uptake of DNA directly from the surrounding environment. Some bacteria are naturally competent and internalize DNA that can be integrated to their genetic patrimony by mechanism such as recombination. (**d**) Vesicle mediated gene transfer is the newer mechanism found to be involved in HGT. It involves formation of vesicles that may include plasmidial or fragments of chromosomal DNA. The vesicles fuse to membranes and fade into recipient cells. Vesicle-transported DNA can be a plasmid or a fragment of DNA that might recombine with the chromosome of recipient bacterial cells. Vesicle mediated HGT was extensively reviewed by Domingues and Nielsen 2017 [328]. Figures were drawn from scratch using Freehands 10 (Macromedia, San Francisco, CA, USA).

**Figure 7 molecules-26-07153-f007:**
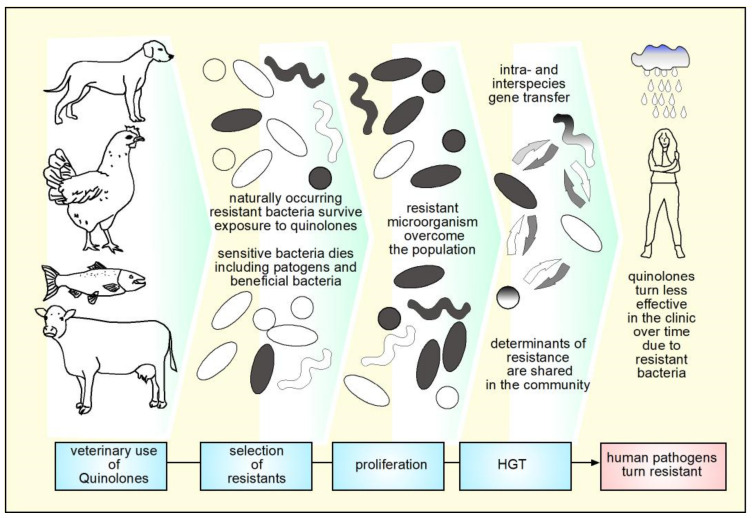
The use of quinolones in veterinary medicine increases selective pressure and emergence of resistance. Once antibiotics-resistant bacteria are increasing in fraction of bacterial populations, the genetic resistance markers can be horizontally transferred to recipient bacteria of the same or different species or genus. Therefore, the use of antimicrobials in the family of quinolones risks the future of quinolones as effective antimicrobials for human use. It is important to withdraw quinolones from use in animal production to release selective pressure. Figures were drawn from scratch using Freehands 10 (Macromedia, San Francisco, CA, USA).

**Figure 8 molecules-26-07153-f008:**
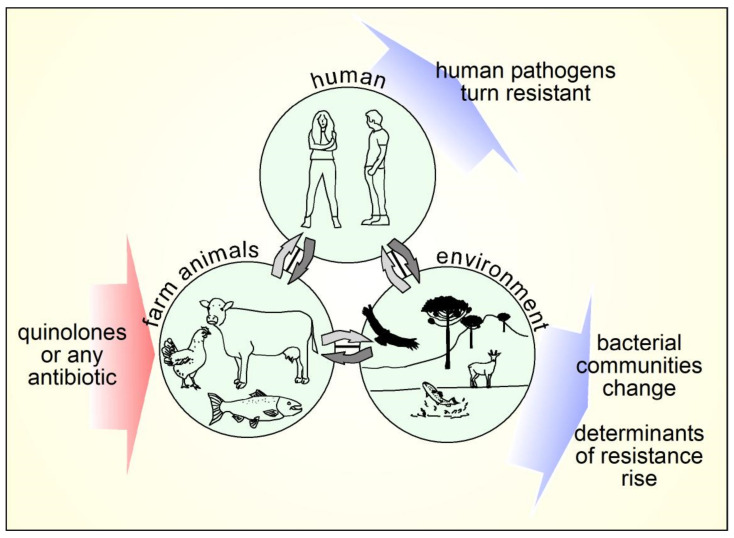
The effects of quinolones, and antibiotics, when used in farm animals. The use in animal production puts selective pressure with consequences on the environment and humans. The same effects will happen if the antibiotics are applied at the humans or environmental levels. The reasonable use of antibiotics intended for humans should be a priority and without question, those antibiotics should be banned from massive use, such as use in animal farms. Figures were drawn from scratch using Freehands 10 (Macromedia, San Francisco, CA, USA).

**Table 1 molecules-26-07153-t001:** Compounds that potentiate the antimicrobial effects of quinolones.

Agents (with Antimicrobial Effect)	Relevant Effects	Reference
Feluric acid	Induces oxidative stress. Antimicrobial activity for Gram −/+	[23,24]
CdTe-2.4	Cadmium telluride quantum dots that after excitation produces superoxide	[25]
Kaempferol glycosides	Potentiate quinolones effects in MRSA and vancomycin resistant *S. aureus*	[30]
**(without Antimicrobial Effect)**		
vitamin K_3_	Inhibition and decreased transcription of efflux pump NorA	[27]
IMP-1700	Inhibition of DNA-repair systems of bacteria	[28]
biochanin A derivatives	Inhibition of Efflux pumps non-tuberculosis *Mycobacterium*	[31]
L-serine	Enhances quinolones effects by increasing oxidative stress	[32]
Palmatine	Restores antimicrobial activity of ciprofloxacin in quinolones resistant *E. coli*	[33]
Ambroxol	Ambroxol combined with ciprofloxacin presented potent activity at clinical used concentrations against SARS-CoV-2 measured as qPCR of its genome, survival, and lysis plaque formation	[34]

**Table 2 molecules-26-07153-t002:** Effects of quinolones and related molecules, different from anti-bacterials.

Agent	Effect	Reference
**(As Anti-Parasitic)**		
Fluorinated quinone amides	Antiparasitic, anti-trypanosomal	[122]
Metal complexes with FQs	Antiparasitic	[92]
Methyl quinolones-derivates	Antiparasitic	[123]
**(As Anti-Viral)**		
Enrofloxacin, ofloxacin, orbifloxacin, iprofloxacin	Activity anti-HCV	[103]
Fluoroquinolone-isatin hybrids	Anti-HIV and anti-HCV activities	[124]
Quinolones derivates molecules	HIV-1 integrase inhibitors	[125]
**(As Anti-Fungal)**		
Quinolone derivates molecules	Anti-fungal activity against *A. flavus*	[98]
Clinafloxacin triazole hybrids	Anti-fungal activity against *C. albicans* and *C. mycoderma*	[94]
**(As Anti-Inflammatory)**		
Ciprofloxacin Levofloxacin	Anti-inflammatory. Inhibition of microglia inflammatory response	[110]
Moxifloxacin	Immunomodulatory activity. Reduces secretion of IL-α and TNF-α. Anti-inflammatory. Inhibited ERK1/2, p38 and p65-NFκB. Inhibits etoposide pro-inflammatory effects	[111,112,113]
**(As Anti-Cancer/Anti-Tumor)**		
Moxifloxacin	Activity anti-tumor (Cooper complexes)	[119]
Ciprofloxacin and other FQs	Cell cycle arrest and induction of apoptosis	[115,116,117]
Ruthenium complex with quinolones	Anti-cancer	[126]
**(With Other Effects)**		
Enrofloxacin, ofloxacin, orbifloxacin, ciprofloxacin	Inhibitory effects on CYP3A in Dogs	[127]
Ciprofloxacin and enoxacin	Competitive inhibitory of CYP1A2	[78]
**(Quinolones-Related Molecules)**		
Quinazolinones related compounds	Anti-convulsant	[128]
Quinolines	Neuroprotective, Diuretic activity	[129,130]

**Table 3 molecules-26-07153-t003:** Comparative usage of quinolones in food-producing animals during 2013.

Country or Zone	Tons of FQs Used in Animal Production	Tons of Total Antimicrobials for Food-Producing Animals	Percentage	Ref.
China	22,210	78,100	28.4	[239]
Europe	188 ^1^	8060	2.3	[267]
USA	15	9196	0.2	[258]

^1^ includes quinolones and fluoroquinolones.

## Data Availability

No new data were created or analyzed in this study.

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
