# Peer review of "Biological Effects of Quinolones: A Family of Broad-Spectrum Antimicrobial Agents"

_molecules, 2021, doi:10.3390/molecules26237153_

Round 1

Reviewer 1 Report

This is quite interesting of review-type paper about biological effects of quinolones. Such papers are very helpful material for scientist because they collect the most important reports in the field. The author cited a lot of recent work (mostly from 2000 to 2020) on biological properties quinolones, which are building blocks of some important antimicrobial agents, which are still in clinical use. The increasing resistance of bacteria against antibiotics and the reasons why it occurs belong to one of the most challenging scientific problems to be solved. The paper is written and organized well. However, it would appreciate if there were one short subchapter about physico-chemical properties of quinolones. Besides that, this paper requires some minor additional information. For example:

- in my opinion the are no structures in Fig1, but rather chemical formulas of quinolones

- in Figs 4-6. captions should be clearly specified whether these are original arts or adopted from the other sources with the journals permissions,

- the subchapter no. 10 has been written in a telegraphic way. A reader might have an impression that the authors did not have any idea or did not bother enough. There is no new information which cannot be read in the previous parts of the manuscript. I would recommend to re-write it and point out the future of quinolones, if there is any.

Generally, this is a valuable scientific material, and valid for publication in Molecules after minor revision.

Author Response

Please find below the answers to questions and suggestions.

This is quite interesting of review-type paper about biological effects of quinolones. Such papers are very helpful material for scientist because they collect the most important reports in the field. The author cited a lot of recent work (mostly from 2000 to 2020) on biological properties quinolones, which are building blocks of some important antimicrobial agents, which are still in clinical use. The increasing resistance of bacteria against antibiotics and the reasons why it occurs belong to one of the most challenging scientific problems to be solved. The paper is written and organized well. However, it would appreciate if there were one short subchapter about physico-chemical properties of quinolones. Besides that, this paper requires some minor additional information. For example:

- in my opinion the are no structures in Fig1, but rather chemical formulas of quinolones

We changed structure with chemical formula both in the captions of figures and within the text.

- in Figs 4-6. captions should be clearly specified whether these are original arts or adopted from the other sources with the journals permissions,

All figures presented here were elaborated from scratch. We are including more details in programs used to draw chemical formulae and sketching figures.

- the subchapter no. 10 has been written in a telegraphic way. A reader might have an impression that the authors did not have any idea or did not bother enough. There is no new information which cannot be read in the previous parts of the manuscript. I would recommend to re-write it and point out the future of quinolones, if there is any.

We have extended discussion about future of quinolones.

Generally, this is a valuable scientific material, and valid for publication in Molecules after minor revision.

We respectfully thank your revision. We are bringing out a new version that try to conciliate recommendations from the three reviewers

Reviewer 2 Report

Dear editor, thank you very much for your consideration as a reviewer. 

The authors' intent to write a bibliographic review about the antimicrobial activity of quinolines compounds and derivatives.  However, if you review the specialized bibliography in the last 10 years, more than 5 reviews might be found. Even the selected theme is interesting and has enough soundness and scientific appeal, I am afraid the author must improve the manuscript in order to present some differences from the review reported before. So, I strongly suggest as follow:

  1. The title must change to "Quinolones: a family of broad-spectrum antimicrobial agents" due to the authors focused just on antimicrobial activities.
  2. Sorting the structure according to action mechanism/specific target
  3.  Sar or QSAR of the compounds.
  4.  Synthetic and natural quinolines.
  5. Any computing-aided design and synthesis of quinolines. 

I am confident the authors can address these suggestions, and send back the manuscript. 

My sincerely, 

Author Response

Please find below the answers to questions and suggestions

The authors' intent to write a bibliographic review about the antimicrobial activity of quinolines compounds and derivatives.  However, if you review the specialized bibliography in the last 10 years, more than 5 reviews might be found. Even the selected theme is interesting and has enough soundness and scientific appeal, I am afraid the author must improve the manuscript in order to present some differences from the review reported before. So, I strongly suggest as follow:

  1. The title must change to "Quinolones: a family of broad-spectrum antimicrobial agents" due to the authors focused just on antimicrobial activities.

We agree with reviewer 2 and have changed the title accordingly

  1. Sorting the structure according to action mechanism/specific target.

The overall structure of the manuscript relies more into chronologically sorting molecules rather than sorting according to structure v/s mechanisms/target. There are instances, such as chapter 3 “Physicochemical properties of quinolones” and chapter 5 “Structure-activity relationship of quinolones” were structure, or at list functional groups in the quinolone formulae, are related to special features, potency, spectrum, solubility, penetration though tissue, etc. also, modifications to quinolones core that produced undesired effects are discussed.

  1.  Sar or QSAR of the compounds.

We have included a new chapter about structure-activity relationship (SAR and Quantitative SAR, QSAR). We have reviewed functional and structural features that influences antimicrobial potency, spectrum, penetration into tissues, but also modifications that introduced undesired effects. Because there is little information about computing-aided design, we discussed in this chapter some few topics where computing design and modeling has been helpful to predict desired features of quinolones.

  1.  Synthetic and natural quinolines.

Since the study of natural occurring quinolones remains mostly in the structural field and preclinical stage, it is difficult to build a robust chapter. Indeed, the manuscript already includes available information regarding sources, structures, and antimicrobial activities of natural quinolones. This chapter remains after physic-chemistry of synthetic quinolones and before the SAR chapter.

  1. Any computing-aided design and synthesis of quinolines.

Since few reports are published about computing-aided design of quinolones and that we are not expert in either computing-aided modeling field, or in chemical synthesis, we included this information in the SAR chapter.

I am confident the authors can address these suggestions, and send back the manuscript. 

We appreciate corrections and suggestions by Reviewer 2, however we have to bring out a version of this manuscript that conciliate the suggestions from all three referees.

Reviewer 3 Report

Millanao and co-authors have provided a well referenced review on the biological effects of quinolone antibiotics. In addition to antimicrobial properties of these compounds, biological activity of quinolone and FQ activity against other microorganisms and indications is discussed. This also includes an informative section on the environmental and human impacts of quinolone and FQ use in veterinary settings. This is an informative manuscript which is suitable for publication in Molecules with some revisions.

Major revisions:

I believe this report would benefit from the introduction of additional figures and/or more detail within the current figures. The authors should also consider readers who are do not specialise in quinolones antimicrobials. For each chemical compound discussed a struct should be included. This can also be helpful to demonstrate the chemical development of this series without specifically discussing that aspect. Figure 1 should be expanded to include this information as well as expanding on the different generations which is discussed at a later stage in the review. Again, showing the structures with help to clarify what develops have occurred between each generation. As the representation of potency in this figure is a little arbitrary and non-scientific, I would question its relevance.

A structure should also be include which includes the numbering of the quinolone core. This helps clarify discussion of positions through-out the text (e.g. L412).

More information is required in the captions of other schematics - for example Figure 4 and 6. It would be worthwhile including more detail in the captions so that the figures can be interpreted when viewed in isolation as well as when reading the text. Figures are an opportunity to summarise and display key information.

Minor revisions:

The article is generally well written but there are a number of typographical and grammatical errors. The latter is predominantly in the form of misused plurals. Some examples of typographical errors or grammar are below:

L19 - Well distribution...

L22 - change 'leaving quinolones family' to 'leaving the quinolone family'

L25 - change 'antimicrobials' to 'antimicrobial'

L51 - change 'routinary' to 'routine'

L79 - change 'readying' to 'reading'

L82 -change 'up-to-date' to 'to-date'

L89 - 'Perhaps its oral efficacy' this sentence is not clear.

L92 - change 'sixth' to six'

L98 - 'up-to-day' rephrase.

L140 - change 'broad' to 'broaden'

L150 - 'was early described' 

L199 - change 'which' to 'with'

L207 - 'develop' to 'developed'

L220 - 'logic' to 'logical'

L233 - 'multiples' to 'multiple'

L235 - 'compound' to 'compounds'

L243 - 'that' to 'than'

L310 - fluroquinolones is mispelled.

L458 - 'this inhibitor' or 'these inhibitors'

L520 - remove 'Up'

L535 - change 'regulate' to 'regulates'

L550 - 'this changes result' should be these?

L803 - 'released' to 'release'

L852 - Mg+2 and Ca+2 should be Mg2+ and Ca2+

L955 - 'treat' to 'threat'

L956 - 'not' to 'no'

L1000 - 'have shown effective' is not clear

L1010 - 'de' to 'the'

This is not an extensive list; I recommend a further proof-read.

Please use full names prior to the first use of an abbreviation e.g. L520.

Author Response

Please find below the answers to questions and suggestions

Millanao and co-authors have provided a well referenced review on the biological effects of quinolone antibiotics. In addition to antimicrobial properties of these compounds, biological activity of quinolone and FQ activity against other microorganisms and indications is discussed. This also includes an informative section on the environmental and human impacts of quinolone and FQ use in veterinary settings. This is an informative manuscript which is suitable for publication in Molecules with some revisions.

MAJOR REVISIONS: I believe this report would benefit from the introduction of additional figures and/or more detail within the current figures. The authors should also consider readers who are do not specialise in quinolones antimicrobials. For each chemical compound discussed a struct should be included. This can also be helpful to demonstrate the chemical development of this series without specifically discussing that aspect. Figure 1 should be expanded to include this information as well as expanding on the different generations which is discussed at a later stage in the review. Again, showing the structures with help to clarify what develops have occurred between each generation. As the representation of potency in this figure is a little arbitrary and non-scientific, I would question its relevance.

Response 1

About including more figures and details into existing figures: we have included a new figure depicting up-to-date mechanisms of horizonal gene transfer between bacteria. Also, we are including a new figure that includes formulae of several quinolones discussed within the text. In addition, some figures have been colored to denotate numbers and other details.

Representation of quinolones potency in figure one: We agree that representation of potency was totally arbitrary and may imprint a slopy impression on readers.  That part of figure 1 was changed by a continuous gradient from early through newer quinolones representing the overall qualitative increase of potency.

A structure should also be include which includes the numbering of the quinolone core. This helps clarify discussion of positions through-out the text (e.g. L412).

Response 2: numbering of quinolones core in figure 1 has been included. We also included the numbering in nalidixic acid formula in the new Figure 3.

More information is required in the captions of other schematics - for example Figure 4 and 6. It would be worthwhile including more detail in the captions so that the figures can be interpreted when viewed in isolation as well as when reading the text. Figures are an opportunity to summarise and display key information.

Response 3: more in detail information has been added in captions, especially for old figures 4 through 6 (now 6 through 8)

MINOR REVISIONS:The article is generally well written but there are a number of typographical and grammatical errors. The latter is predominantly in the form of misused plurals. Some examples of typographical errors or grammar are below:

Response 3:

We thank all the many typographical and grammatical corrections, the full list provided by reviewer 3 was addressed. In addition, we have done further proofread of the text, found, and corrected many other minor errors.

L19 - Well distribution...             corrected

L22 - change 'leaving quinolones family' to 'leaving the quinolone family'              corrected

L25 - change 'antimicrobials' to 'antimicrobial'     corrected

L51 - change 'routinary' to 'routine'         corrected

L79 - change 'readying' to 'reading         corrected

L82 -change 'up-to-date' to 'to-date'        corrected

L89 - 'Perhaps its oral efficacy' this sentence is not clear.            Addressed

L92 - change 'sixth' to six'          corrected

L98 - 'up-to-day' rephrase.         corrected

L140 - change 'broad' to 'broaden'          corrected

L150 - 'was early described'       corrected

L199 - change 'which' to 'with'    corrected

L207 - 'develop' to 'developed'   corrected

L220 - 'logic' to 'logical'              corrected

L233 - 'multiples' to 'multiple'      corrected

L235 - 'compound' to 'compounds'         corrected

L243 - 'that' to 'than'       corrected

L310 - fluroquinolones is mispelled.        corrected

L458 - 'this inhibitor' or 'these inhibitors'              corrected

L520 - remove 'Up'        corrected

L535 - change 'regulate' to 'regulates'                 corrected

L550 - 'this changes result' should be these?      corrected

L803 - 'released' to 'release'       corrected

L852 - Mg+2 and Ca+2 should be Mg2+ and Ca2+          corrected

L955 - 'treat' to 'threat'   corrected

L956 - 'not' to 'no'                                                          corrected

L1000 - 'have shown effective' is not clear           addressed

L1010 - 'de' to 'the'                    corrected

This is not an extensive list; I recommend a further proof-read.    corrected

Please use full names prior to the first use of an abbreviation e.g. L520.   corrected

Round 2

Reviewer 2 Report

Thank you very much for addressing all of the suggestions. I have no doubt a lot of research will cite this manuscript in the short future. Excellent contribution!